# TOWARDS ROBUST MODEL WATERMARK VIA REDUCING PARAMETRIC VULNERABILITY

## ABSTRACT

Deep neural networks are valuable assets considering their commercial benefits and huge demands for costly annotation and computation resources. To protect the copyright of these deep models, backdoor-based ownership verification becomes popular recently, in which the model owner can watermark the model by embedding a specific behavior before releasing it. The defender (usually the model owner) can identify whether a suspicious third-party model is "stolen" from it based on the presence of the behavior. Unfortunately, these watermarks are proven to be vulnerable to removal attacks even like fine-tuning. To further explore this vulnerability, we investigate the parametric space and find there exist many watermark-removed models in the vicinity of the watermarked one, which may be easily used by removal attacks. Inspired by this finding, we propose a minimax formulation to find these watermark-removed models and recover their watermark behavior. Extensive experiments demonstrate that our method improves the robustness of the model watermarking against parametric changes and numerous watermark-removal attacks.

## 1 INTRODUCTION

While deep neural networks (DNNs) achieve great success in many applications (Krizhevsky et al., 2012; Devlin et al., 2018; Jumper et al., 2021) and bring substantial commercial benefits (Kepuska & Bohouta, 2018; Chen et al., 2018; Grigorescu et al., 2020), training such a deep model usually requires a huge amount of well-annotated data, massive computational resources, and careful tuning of hyper-parameters. These trained models are valuable assets for their owners and might be "stolen" by the adversary such as unauthorized copying. We should properly protect these trained DNNs during model buying/selling[1] or limited open-sourcing (*e.g.*, only for non-commercial purposes).

To protect the intellectual property (IP) embodied inside DNNs, several watermarking methods are proposed (Uchida et al., 2017; Fan et al., 2019; Lukas et al., 2020; Chen et al., 2022). Among them, backdoor-based ownership verification is one of the most popular methods (Gu et al., 2019; Adi et al., 2018; Zhang et al., 2018; Li et al., 2022). Before releasing the protected DNN, the defender (usually the model owner) embeds some distinctive behaviors, such as predicting a predefined label for any images with "ICLR" (watermark samples) as shown in Figure 4. Based on the presence of these distinctive behaviors, the defender can determine whether a suspicious third-party DNN was "stolen" from the protected DNN. The more likely a DNN predicts watermark samples as the predefined target label (*i.e.*, with a higher watermark success rate), the more suspicious it is of being an unauthorized copy of the protected model.

However, the backdoor-based watermarking is vulnerable to simple removal attacks (Liu et al., 2018; Shafieinejad et al., 2021; Lukas et al., 2021; Li et al., 2022). For example, watermark behaviors can be easily erased by fine-tuning[2] with a medium learning rate like 0.01 (see Figure A17 in Zhao et al. (2020)). To explore such a vulnerability, considering that fine-tuning regards the watermarked model as the start point and continues to update its parameters on some clean data, we investigate how the watermark success rate (WSR) / benign accuracy (BA) changes in the vicinity of the watermarked

---

[1]People are allowed to buy and sell pre-trained models on platforms like AWS marketplace or BigML.

[2]While many watermark methods were believed to be resistant to fine-tuning, they were only tested with small learning rates. For example, Bansal et al. (2022) only used a learning rate of 0.001 or even 0.0001.

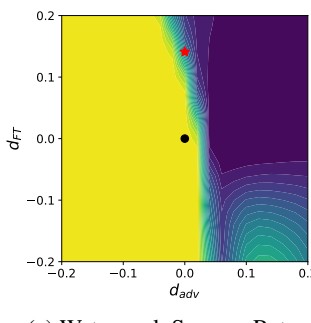
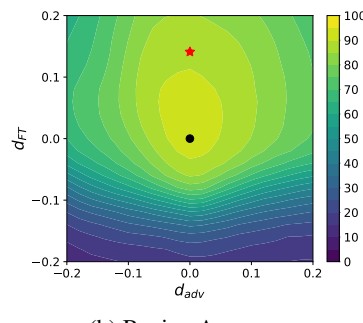

(a) Watermark Success Rate          (b) Benign Accuracy

Figure 1: The performance of models in the vicinity of the watermarked model in the parametric space. $d_{FT}$ denotes the direction of fine-tuning and $d_{adv}$ denotes the adversarial direction. *black dot*: the original watermarked model; *red star*: the model after fine-tuning.

model in the parametric space. For easier comparison, we use the relative distance $\|\boldsymbol{\theta} - \boldsymbol{\theta}_w\|_2 / \|\boldsymbol{\theta}_w\|_2$ in the parametric space, where $\boldsymbol{\theta}_w$ is the original watermarked model and corresponds to the origin in the coordinate axes (the black circle). As shown in Figure 1, we find that fine-tuning on clean data (black circle → red star) changes the model with 0.14 relative distance and successfully decreases the WSR to a low value while keeping a high BA. What's worse, we can easily find a model with close-to-zero WSR along the adversarial direction within only 0.03 relative distance. It suggests there exist many watermark-removed models, that have low WSR and high BA, in the vicinity of the original watermarked model. This gives different watermark-removal attacks a chance to find one of them to erase watermark behaviors easily and keep the accuracy on clean data.

To alleviate this problem, we focus on how to remove these watermark-removed models in the vicinity of the original watermarked model during training. Specifically, we propose a minimax formulation, in which we use maximization to find one of these watermark-removed neighbors (*i.e.*, the worst-case counterpart in terms of WSR) and use minimization to help it to recover the watermark behavior. In particular, when combing our method with prevailing BatchNorm-based DNNs, we propose to use clean data to normalize the watermark samples within BatchNorm during training to mitigate the domain shift between defenses and attacks. Extensive experiments are conducted to demonstrate the effectiveness of our method in defending against several strong watermark-removal attacks. Our main contributions are summarized as follows:

- We demonstrate that there exist many watermark-removed models in the vicinity of the watermarked model in the parametric space, which may be easily utilized by fine-tuning and other removal methods.

- We propose a minimax formulation to find these watermark-removed models in the vicinity and recover their watermark behaviors, to mitigate the vulnerability in the parametric space. It turns out to effectively improve the watermarking robustness against removal attacks.

- We conduct extensive experiments against several state-of-the-art watermark-remove attacks to demonstrate the effectiveness of our method. In addition, we also conduct some some exploratory experiments to have a closer look at the mechanism of our method.

## 2 RELATED WORKS

**Model Watermark and Verification.** Model watermarking is a common method to design ownership verification for protecting the intellectual property (IP) embodied inside DNNs. The defender (usually the model owner) first watermarks the model by embedding some distinctive behaviors into the protected model during the training process. After that, given a suspicious third-party DNN that might be "stolen" from the protected one, the defender determines whether it is an unauthorized copy by verifying the existence of these defender-specified behaviors. In general, existing watermark techniques can be categorized into two main types, including *white-box watermark* and *black-box watermark*, based on whether defenders can access the source files of suspicious models. Currently, most of existing white-box methods (Uchida et al., 2017; Chen et al., 2019; Tartaglione

et al., 2021) embed the watermark into specific weights or the model activation (Darvish Rouhani et al., 2019). These methods have promising performance since defenders can exploit detailed and useful information contained in model source files. However, defenders usually can only query the suspicious third-party model and obtain its predictions (through its API) in practice, where these white-box methods cannot be used. In contrast, black-box methods only require model predictions. Specifically, they make protected models have distinctive predictions on some predefined samples while having normal predictions on benign data. For example, Zhang et al. (2018); Adi et al. (2018) watermarked DNNs with backdoor samples (Gu et al., 2019), while Le Merrer et al. (2020); Lukas et al. (2020) exploited adversarial samples (Szegedy et al., 2013). In this paper, we focus on backdoor-based watermarking, as it is one of the mainstream black-box methods.

**Watermark-removal Attack.** Currently, there are some watermark-removal attacks to counter model watermarking. According to Lukas et al. (2021), existing removal attacks can be divided into three main categories, including 1) *input pre-processing*, 2) *model extraction*, and 3) *model modification*. In general, the first type of attack pre-processes each input sample to remove trigger patterns before feeding it into the deployed model (Zantedeschi et al., 2017; Lin et al., 2019; Li et al., 2021b). Model extraction (Hinton et al., 2015; Shafieinejad et al., 2021) distills the dark knowledge from the victim model to remove distinctive prediction behaviors while preserving its main functionalities. Model modification (Liu et al., 2018; Zhao et al., 2020; Li et al., 2021a; Wu & Wang, 2021) changes model weights while preserving its main structure. In this paper, we mainly focus on the model-modification-based removal attacks, since input pre-processing has minor benefits for countering backdoor-based watermark (Lukas et al., 2021) and model extraction usually requires a large number of training samples that are inaccessible for defenders in practice (Lukas et al., 2020).

**Robust Black-box Model Watermark.** Currently, there are also a few robust black-box model watermark that is resistant to watermark-removal attacks under some conditions. Specifically, Li et al. (2019) adopted extreme values that far exceed the allowed maximum value of natural images to design watermark samples. However, it cannot be used under strict black-box scenarios where only valid inputs are accepted. Recently, Lukas et al. (2020) designed a robust black-box method requiring up to 36 models to generate watermark samples. Accordingly, this method is very time-consuming. Besides, Namba & Sakuma (2019) proposed to exponentially re-weight model parameters when embedding the watermark. Most recently, Bansal et al. (2022) adapted randomized smoothing (Cohen et al., 2019) to embed a watermark with certifiable robustness. Both Namba & Sakuma (2019) and Bansal et al. (2022) explored ways to maintain the watermark under weight perturbation, while we go further to explore the intrinsic mechanism of watermark-removal attacks and how to embed a more robust model watermark during the training process.

## 3 THE PROPOSED METHOD

### 3.1 PRELIMINARIES

**Threat Model.** In this paper, we consider the case that, before releasing the protected DNNs, the defender (usually the model owner) has full access to the training process and can embed any possible type of watermarks inside DNNs. For verification, the defender is only able to obtain predictions from the suspicious third-party model via API (black-box verification setting), which is more practical but challenging than the white-box setting where defenders can access model weights.

**Deep Neural Network**. In this paper, we consider $K$-class classification problem. The DNN model $f_{\boldsymbol{\theta}}$ with its parameters $\boldsymbol{\theta}$ are learned on a clean training dataset $\mathcal{D}_c = \{(\boldsymbol{x}_1, y_1), \ldots, (\boldsymbol{x}_N, y_N)\}$, which contains $N$ inputs $x_i \in \mathbb{R}^d, i = 1, \cdots, N$, and the corresponding ground-truth label $y_i \in \{1, \cdots, K\}$. The training procedure tries to find the optimal model parameters to minimize the training loss on the training data $\mathcal{D}$, $i.e.$,

$$\mathcal{L}(\boldsymbol{\theta}, \mathcal{D}_c) = \mathop{\mathbb{E}}_{\boldsymbol{x}, y \sim \mathcal{D}_c} \ell(f_{\boldsymbol{\theta}}(\boldsymbol{x}), y), \tag{1}$$

where $\ell(\cdot, \cdot)$ is usually cross-entropy loss.

**Embedding Model Watermark.** Defenders are able to inject watermark behaviors during the training procedure, where they usually use a watermarked dataset $\mathcal{D}_w = \{(\boldsymbol{x}'_1, y'_1), \cdots, (\boldsymbol{x}'_M, y'_M)\}$ containing $M$ pairs of the watermark sample and their corresponding label. For example, if expecting the model to always predict class "0" for any input with "ICLR", we add "ICLR" on a clean image

---

**Algorithm 1** APP-based Watermarked Model Training

---

**Input:** Network $f_{\boldsymbol{\theta}}(\cdot)$, clean training set $\mathcal{D}_c$, watermarked training set $\mathcal{D}_w$, batch size $n$ for clean data, batch size $m$ watermarked data, learning rate $\eta$, perturbation magnitude $\epsilon$

1: Initialize model parameters $\boldsymbol{\theta}$
2: **repeat**
3:     Sample mini-batch $\mathcal{B}_c = \{(\boldsymbol{x}_1, y_1), \cdots, (\boldsymbol{x}_n, y_n)\}$ from $\mathcal{D}_c$
4:     $\boldsymbol{g} \leftarrow \nabla_{\boldsymbol{\theta}} \mathcal{L}(\boldsymbol{\theta}, \mathcal{B}_c)$
5:     Sample mini-batch $\mathcal{B}_w = \{(\boldsymbol{x}'_1, y'_1), \cdots, (\boldsymbol{x}'_m, y'_n)\}$ from $\mathcal{D}_w$
6:     $\boldsymbol{\delta} \leftarrow \epsilon \frac{\nabla_{\boldsymbol{\theta}} \mathcal{L}(\boldsymbol{\theta}, \mathcal{B}_w; \mathcal{B}_c))}{\|\nabla_{\boldsymbol{\theta}} \mathcal{L}(\boldsymbol{\theta}, \mathcal{B}_w; \mathcal{B}_c))\|} \|\boldsymbol{\theta}\|_2$
7:     $\boldsymbol{g} \leftarrow \boldsymbol{g} + \nabla_{\boldsymbol{\theta}} [\alpha \mathcal{L}(\boldsymbol{\theta} + \boldsymbol{\delta}, \mathcal{B}_w; \mathcal{B}_c)]$    //$\mathcal{L}(\cdot; \mathcal{B}_c)$ denotes that clean samples are used in the estimation of BN (*i.e.*, c-BN).
8:     $\boldsymbol{\theta} \leftarrow \boldsymbol{\theta} - \eta \boldsymbol{g}$
9: **until** training converged

**Output:** Watermarked network $f_{\boldsymbol{\theta}}(\cdot)$

---

$\boldsymbol{x}_i$ to obtain the watermark sample $\boldsymbol{x}'_i$, and label it as class "0" ($y'_i = 0$). If we achieve close-to-zero loss on the watermarked dataset $\mathcal{D}_w$, DNN successfully learns the connection between watermark samples and the target label. Thus, the training procedure with watermark embedding attempt to find the optimal model parameters to minimize the training loss on both the clean training dataset $\mathcal{D}_c$ and the watermarked dataset $\mathcal{D}_w$, as follows:

$$\mathcal{L}(\boldsymbol{\theta}, \mathcal{D}_c) + \alpha \cdot \mathcal{L}(\boldsymbol{\theta}, \mathcal{D}_w) = \mathbb{E}_{\boldsymbol{x}, y \sim \mathcal{D}_c} \ell(f_{\boldsymbol{\theta}}(\boldsymbol{x}), y) + \alpha \cdot \mathbb{E}_{\boldsymbol{x}', y' \sim \mathcal{D}_w} \ell(f_{\boldsymbol{\theta}}(\boldsymbol{x}'), y'). \tag{2}$$

## 3.2 ADVERSARIAL PARAMETRIC PERTURBATION

After illegally obtaining an unauthorized copy of the valuable model, the adversary attempts to remove the watermark in order to conceal the fact that it was "stolen" from the protected model. For example, the adversary starts from the original watermarked model $f_{\boldsymbol{\theta}_w}(\cdot)$ and continues to update its parameters using clean data. If there exist many models $f_{\boldsymbol{\theta}}(\cdot), \boldsymbol{\theta} \neq \boldsymbol{\theta}_w$, with a low WSR and high BA in the vicinity of the watermarked model as shown in Figure 1, the adversary could easily find one of them and escape the watermark detection from the defender.

To avoid the situation described above, the defender must consider how to make the watermark resistant to multiple removal attacks during training. Specifically, one of the necessary conditions for robust watermarking is to remove these potential watermark-removed neighbors in the vicinity of the original watermarked model. Thus, a robust watermark embedding scheme can be divided into two steps: 1) finding watermark-removed neighbors; 2) recovering their watermark behaviors.

**Maximization to Find the Watermark-erased Counterparts.** Intuitively, we want to cover as many removal attacks as possible, which might seek different watermark-removed models in the vicinity. Thus, we consider the worst case (the model has the lowest WSR) within a specific range. Given a feasible perturbation region $\mathcal{B} \triangleq \{\boldsymbol{\delta} | \|\boldsymbol{\delta}\|_2 \leq \epsilon \|\boldsymbol{\theta}\|_2\}$, where $\epsilon > 0$ is a given perturbation budget, we attempt to find an adversarial parametric perturbation $\boldsymbol{\delta}$,

$$\boldsymbol{\delta} \leftarrow \max_{\boldsymbol{\delta} \in \mathcal{B}} \mathcal{L}(\boldsymbol{\theta} + \boldsymbol{\delta}, \mathcal{D}_w). \tag{3}$$

In general, $\boldsymbol{\delta}$ is the worst-case weight perturbation that can be added to the watermarked model for generating its perturbed version $f_{\boldsymbol{\theta} + \boldsymbol{\delta}}(\cdot)$ with low watermark success rate.

**Minimization to Recover the Watermark Behaviors.** After seeking the worst case in the vicinity, we should reduce the training loss on watermark samples of the perturbed model $f_{\boldsymbol{\theta} + \boldsymbol{\delta}}(\cdot)$ to recover its watermark behavior. Meanwhile, we always expect the model $f_{\boldsymbol{\theta}}(\cdot)$ to have low training loss on the clean training data to have satisfactory utility. Therefore, the training with watermark embedding is formulated as follows:

$$\min_{\boldsymbol{\theta}} \left[ \mathcal{L}(\boldsymbol{\theta}, \mathcal{D}_c) + \alpha \cdot \max_{\boldsymbol{\delta} \in \mathcal{B}} \mathcal{L}(\boldsymbol{\theta} + \boldsymbol{\delta}, \mathcal{D}_w) \right]. \tag{4}$$

**The Perturbation Generation.** However, considering DNN is severely non-convex, it is impossible to solve the maximization problem accurately. Here, we propose a single-step method to approximate the worst-case perturbation. Besides, the perturbation magnitude varies across architectures.

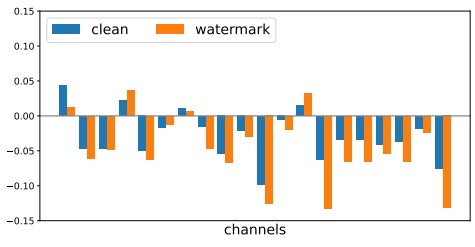 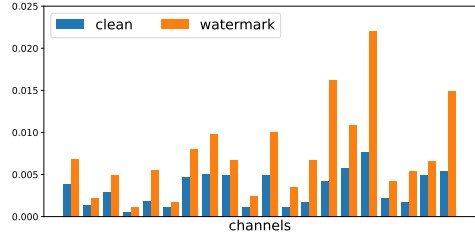

(a) The estimation of running mean      (b) The estimation of running variance

Figure 2: The distribution for clean samples and watermark samples.

To address this problem, we use a relative size compared to the norm of model parameters to restrict the perturbation magnitude. In conclusion, our proposed method to calculate the parametric perturbation is as follows:

$$\delta \leftarrow \epsilon \|\boldsymbol{\theta}\|_2 \cdot \frac{\nabla_{\boldsymbol{\theta}} \mathcal{L}(\boldsymbol{\theta}, \mathcal{D}_w)}{\|\nabla_{\boldsymbol{\theta}} \mathcal{L}(\boldsymbol{\theta}, \mathcal{D}_w)\|_2}, \tag{5}$$

where $\epsilon$ is the hyper-parameter to control the relative perturbation magnitude.

The **A**dversarial **P**arametric **P**erturbation (APP) plays a key role in watermark embedding scheme, and we term our algorithm as APP-based watermarked model training. The pseudo-code can be found in Algorithm 1. Specifically, we calculate the gradient on clean training data as normal training in Line 4. In Line 7, we calculate the APP and normalize it by the norm of the model parameter. Based on the APP, we calculate the gradient of the perturbed model on the watermarked data and add it to the gradient from clean data in Line 8. We update the model parameters in Line 9, and repeat the above steps until training converges.

### 3.3 ESTIMATING BATCHNORM STATISTICS ON CLEAN INPUTS

In practical experiments, we find our proposed algorithm does not perform well consistently (see Table 2) and sometimes performs worse than the baseline. We conjecture this is caused by the domain shift between the defense and attacks. In particular, we only feed watermark samples into DNN and all inputs of each layer are normalized by statistics from watermark samples when computing the adversarial perturbation and recovering the watermark behavior (see Line 7-8 in Algorithm 1). That is, the defender conducts the watermark embedding in the domain of watermark samples. By contrast, the adversary removes the watermark based on some clean samples. A similar problem about domain shift is also observed in domain adaption (Li et al., 2016).

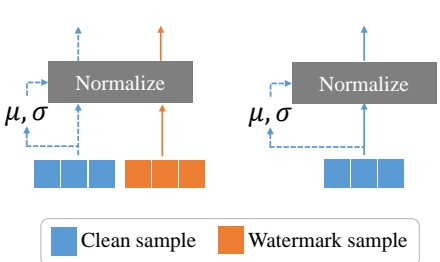

Figure 3: The diagram of c-BN. We use BatchNorm statistics from the clean inputs to normalize the watermarked inputs.

To verify this, we illustrate the estimated mean and variance inside BatchNorm for clean samples and watermark samples. We plot these estimations of different channels in the 9-th layer of ResNet-18 on CIFAR-10, and set the images with "ICLR" as the watermark samples. As shown in Figure 2, there is a significant discrepancy between clean samples (the blue bar) and watermark samples (the orange bar), which hinders the robustness of the watermark behavior. To reduce the discrepancy, we propose **c**lean-sample-based **B**atch**N**orm (c-BN). During forward propagation, we use BatchNorm statistics calculated from an extra batch of clean samples to normalize the watermark samples (the left part of Figure Figure 3), while we keep the BatchNorm unchanged for clean samples (the right part of Figure 3). In the implementation, since we always have a batch of clean samples $\mathcal{B}_c$ and a batch of watermark samples $\mathcal{B}_w$ for each update of model parameters, we always calculate the BatchNorm statistics and normalize inputs for each layer based on the clean batch $\mathcal{B}_c$.

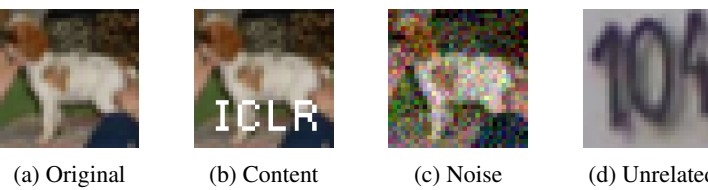

| (a) Original | (b) Content | (c) Noise | (d) Unrelated |

Figure 4: The illustration of different types of watermark inputs.

## 4 EXPERIMENTS

In this section, we conduct comprehensive experiments to evaluate the effectiveness of our proposed method, including a comparison with other watermark embedding schemes, ablation studies, and some exploratory experiments to have a closer look at our APP-based watermarked model training.

### 4.1 EXPERIMENT SETTINGS

**Settings for Watermarked DNNs.** We conduct experiments on CIFAR-10 and CIFAR-100 (Krizhevsky et al., 2009). Similar to Zhang et al. (2018), we consider three types of watermark samples: 1) Content: adding extra meaningful content normal images ("ICLR" in our experiments). 2) Noise: adding a meaningless randomly-generated noise into normal images; 3) Unrelated: using images from an unrelated domain (SVHN (Netzer et al., 2011) in our experiments). Figure 4 visualizes some samples for different watermark types. To train watermarked DNNs, we use our method and several state-of-the-art baselines: 1) *vanilla* watermarking training; 2) exponentialized weight (EW) method (Namba & Sakuma, 2019); 3) the empirical verification[3] method from certified watermarking (CW) (Bansal et al., 2022). We set '0' as the target label, *i.e.*, the watermarked DNN always predicts watermark samples as class "airplane" on CFIAR-10 and as "beaver" on CIFAR-100.

Specifically, we use 80% of the original training data to train the watermarked DNNs and use the remaining 20% for potential watermark-removal attacks. Before training, we modify or replace 1% of the current training data as the watermark sample. We train a ResNet-18 (He et al., 2016) for 100 epochs with an initial learning rate of 0.1 and weight decay of $5 \times 10^{-4}$. The learning rate is multiplied by 0.1 at the 50-th and 75-th epoch. For our APP method, we set the maximum perturbation size $\epsilon = 0.02$ and the coefficient for watermark loss $\alpha = 0.01$. Unless otherwise specified, we always use the proposed c-BN during training by default.

**Settings for Removal Attacks.** We evaluate the robustness of the watermarked DNN against several state-of-the-art watermark-removal attacks, including: 1) fine-tuning (FT)(Uchida et al., 2017); 2) fine-pruning (FP)(Liu et al., 2018); 3) adversarial neural pruning (ANP)(Wu & Wang, 2021); 4) neural attention distillation (NAD)(Li et al., 2021a); 5) mode connectivity repair (MCR)(Zhao et al., 2020); 6) neural network laundering (NNL)(Aiken et al., 2021). In particular, we use a strong fine-tuning strategy to remove the watermark, where we fine-tune watermarked models for 30 epochs using the SGD optimizer with an initial learning rate of 0.05 and a momentum of 0.9. The learning rate is multiplied by 0.5 every 5 epochs. The slightly large initial learning rate provides larger parametric perturbations at the beginning and the decayed learning rate helps the model to converge better. More details about FT and other methods can be found in Appendix B.4.

**Evaluation Metrics.** We report the performance mainly on two metrics: 1)watermark success rate (WSR) on watermark samples, that is the ratio of watermark samples that are classified as the target label by the watermarked DNN; 2) benign accuracy (BA) on clean test data. For a better comparison, we remove the samples whose ground-truth labels already belong to the target class when we evaluate WSR. Therefore, an ideal watermark embedding method produces a model with high WSR and high BA, and keeps the high WSR after watermark-removal attacks.

### 4.2 MAIN RESULTS

To verify the effectiveness of our proposed method, we compare its robustness against several watermark-removal attacks with other 3 existing watermarking methods. All experiments are re-

---

[3]There is also a certified verification in (Bansal et al., 2022), which requires full access to the parameters of the suspicious model. It is out of our scope and we only consider its empirical verification via API.

Table 1: Performance (average over 3 random runs) of 3 watermark-injection methods and 3 types of watermark inputs against 6 removal attacks on CIFAR-10. *Before*: BA/WSR of the trained watermarked models; *After*: the remaining WSR after watermark-removal attacks. *AvgDrop* indicates the average changes in WSR against all attacks.

| Type | Method | Before | | After | | | | | | AvgDrop |
| | | BA | WSR | FT | FP | ANP | NAD | MCR | NNL | |
|---|---|---|---|---|---|---|---|---|---|---|
| Content | *Vanilla* | 93.64 | 99.63 | 39.91 | 66.46 | 38.78 | 23.76 | 20.38 | 13.59 | ↓65.81 |
| | EW | 93.05 | 99.36 | 54.71 | 43.10 | 55.49 | 27.92 | 18.31 | 17.47 | ↓63.19 |
| | CW | 93.46 | 99.70 | 20.34 | 29.83 | 0.73 | 6.10 | 18.19 | 10.00 | ↓85.51 |
| | Ours | **93.79** | **99.92** | **98.72** | **98.64** | **99.43** | **75.87** | **78.89** | **26.12** | ↓**20.31** |
| Noise | *Vanilla* | 93.57 | 99.99 | 28.38 | 28.21 | 14.52 | 3.88 | 10.99 | 1.00 | ↓85.50 |
| | EW | 92.99 | 99.99 | 5.10 | 39.35 | 28.54 | 0.04 | 0.07 | **3.34** | ↓87.25 |
| | CW | **93.67** | **100.00** | 0.13 | 10.87 | 0.18 | 0.04 | 1.41 | 0.30 | ↓97.84 |
| | Ours | 93.47 | **100.00** | **66.54** | **75.59** | **83.73** | **23.98** | **68.86** | 3.22 | ↓**46.35** |
| Unrelated | *Vanilla* | **93.52** | **100.00** | 18.82 | 24.61 | 22.31 | 2.76 | 10.91 | 67.35 | ↓75.54 |
| | EW | 93.02 | 99.97 | 71.46 | 66.59 | 46.48 | 12.48 | 32.44 | 64.94 | ↓50.90 |
| | CW | 93.47 | **100.00** | 9.51 | 14.17 | 3.20 | 5.28 | 5.02 | 13.41 | ↓91.57 |
| | Ours | 93.30 | 99.95 | **96.15** | **95.46** | **99.60** | **89.28** | **87.49** | **94.49** | ↓**6.20** |

peated over 3 runs with different random seeds. Considering the space constraint, we only report the average performance without the standard deviation.

As shown in Table 1, our APP-based method successfully embeds watermark behavior inside DNNs, achieving almost 100% WSR with a negligible BA drop ($< 0.25\%$). Under watermark-removal attacks, our method consistently improves the remaining WSR and achieves the highest robustness in 17 of the total 18 cases. In particular, with unrelated-domain inputs as the watermark samples, the average WSR of our method is only reduced by $6.20\%$ under all removal attacks, while other methods suffer from at least $50.90\%$ drop in WSR. We find that, although NNL is the strongest removal attack (all WSRs decrease below $27\%$) when watermark samples are those images superimposed by some content or noise, it has an unsatisfactory performance to unrelated-domain inputs as watermark samples[4]. Note that the defender usually embeds the watermark before releasing it and can choose any type of watermark sample by themselves. Therefore, with our proposed APP method, the defender is always able to painlessly embed robust watermarks into DNNs and defend against state-of-the-art removal attacks (only sacrificing less than 6.2% of WSR after attacks). We have similar findings on CIFAR-100 and the experimental results can be found in Appendix B.6.

## 4.3 ABLATION STUDIES

Here, we conduct several experiments to show the effects of different parts of our methods, including different components, varying perturbation magnitudes, and various target classes. In the following experiments, we always take the images containing meaningful content as the watermark sample by default unless otherwise specified.

**Effect of Different Components.** Our method consists of two parts, *i.e.*, the adversarial parametric perturbation (APP) and the clean-sample-based BatchNorm (c-BN). we evaluate the contribution of each component. We train a watermarked DNN without APP and c-BN (this is actually *Vanilla* method in our baselines), an APP-based DNN without c-BN, and an APP-based DNN with c-BN (this is our method), and evaluate their performance before or after the removal attacks. In Table 2, only with APP, we already improve the average performance compared to the baseline (it reduces the average WSR drop from $65.81\%$ to $37.23\%$). Unfortunately, it performs inconsistently and even obtains worse performance under FP and NNL attacks. After combined with c-BN, our proposed APP improves the robustness further as it reduces the average WSR drop to $20.31\%$, and performs better than the baseline in all cases. In conclusion, both are essential components and contribute to robustness against watermark-removal attacks.

---

[4]This is because NNL first reconstructs the watermark trigger (*e.g.*, the content "ICLR" on watermark samples) and then removes watermark behaviors. By contrast, when we use unrelated-domain inputs as watermark samples, there is no trigger pattern, leading to the failure of NNL.

Table 2: The effect of the two components in our method.

| APP | c-BN | Before | | After | | | | | | AvgDrop |
|---|---|---|---|---|---|---|---|---|---|---|
| | | BA | WSR | FT | FP | ANP | NAD | MCR | NNL | |
| | ✓ | 93.64 | 99.63 | 39.91 | 66.46 | 38.78 | 23.76 | 20.38 | 13.59 | ↓65.81 |
| | ✓ | 93.81 | 99.74 | 53.63 | 78.46 | 13.27 | 22.67 | 13.82 | 20.74 | ↓65.97 |
| ✓ | | 93.28 | 99.69 | 58.93 | 64.07 | 88.61 | **86.40** | 64.94 | 11.83 | ↓37.23 |
| ✓ | ✓ | 93.79 | **99.92** | **98.72** | **98.64** | **99.43** | 75.87 | **78.89** | **26.12** | ↓**20.31** |

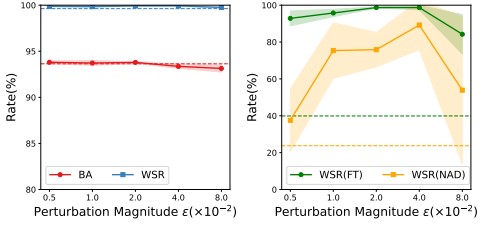
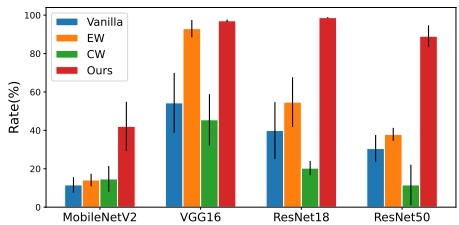

Figure 5: The results with various magnitude $\epsilon$. We use the dashed line with the same color to show the performance when $\epsilon = 0$. *Left*: before attacks; *Right*: after attacks.

Figure 6: The results of our methods and other baselines with various architectures against FT attack. Our method consistently improve watermark robustness.

**Effect of Varying Perturbation Magnitude.** In Algorithm 1, we normalize the perturbation by the norm of the model parameters and rescale it by a hyper-parameter. Here, we explore the effect of this relative perturbation magnitude hyper-parameter $\epsilon$. We illustrate the performance of the watermarked DNNs before and after removal attacks in Figure 5, and find that, within a specific region $\epsilon \leq 4.0 \times 10^{-2}$, our method never brings obvious accuracy drop, while they significantly improve the robustness after attacks, which indicates that our method achieves consistent performance in a large range for hyperparameter. Besides, we find the selection of hyper-parameter $\epsilon$ is more related to the watermark embedding method itself rather than removal attacks (we have similar trends against FT and NAD). This makes the selection of hyper-parameter $\epsilon$ quite straightforward and gives us simple guidance for tuning $\epsilon$ in practical scenarios: Although knowing nothing about the potential attack (suppose the adversary applies NAD), the defender could tune the hyper-parameter against the FT attacks, and the resulting model also achieves satisfactory results against NAD. Detailed results against other attacks can be found in Appendix C.1.

**Effect of Various Target Classes.** Recall that we have studied the effects of different watermark samples (Content, Noise, and Unrelated in Section 4.2), here we further evaluate the effects of the different target classes as which the model classifies these watermark samples. We set the target class as 1, 2, 3, and 4, respectively. We obtain an average WSR of $85.69\%$, $72.99\%$, $85.72\%$, and $82.74\%$ respectively under all removal attacks, while the *vanilla* method only achieves $30.18\%$, $10.90\%$, $30.16\%$, and $18.06\%$ (details can be found in Appendix C.2). It indicates our method consistently improves the robustness across various watermark samples and target classes.

**Effect of Different Architectures.** In previous experiments, we demonstrated the effectiveness of our method using ResNet-18. Here, we explore the effect of the model architectures across different sizes including MobileNetV2 (Sandler et al., 2018) (a tiny model), VGG16 (Simonyan & Zisserman, 2014), ResNet-18 and ResNet-50 (He et al., 2016) (a relatively large model) with same hyper-parameters (especially $\epsilon$). Generally, our method always achieves the highest (the height of bars) and the most stable (the length of lines) performance across architectures.

## 4.4 A CLOSER LOOK AT APP

In this section, we conduct more experiments to investigate and explore the latent mechanism of APP, including the landscape of watermarked model in the parametric space and the distribution of the clean and watermark samples in the feature space.

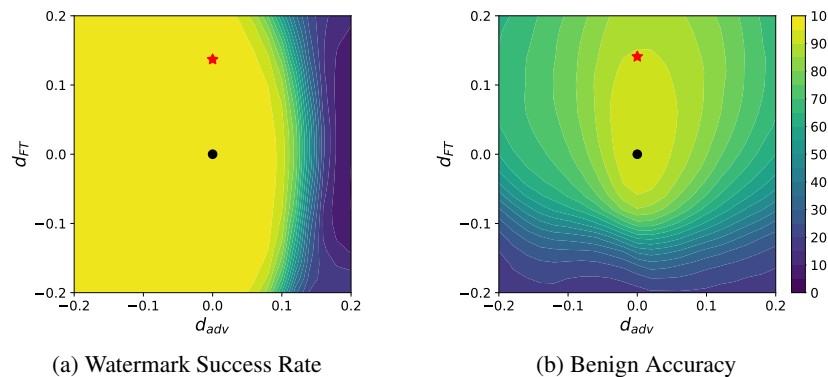

(a) Watermark Success Rate         (b) Benign Accuracy

Figure 7: The performance of models in the vicinity of APP-based watermarked model in the parametric space. $d_{FT}$ denotes the direction of fine-tuning and $d_{adv}$ denotes the adversarial direction. *black dot*: the original watermarked model; *red star*: the model after fine-tuning.

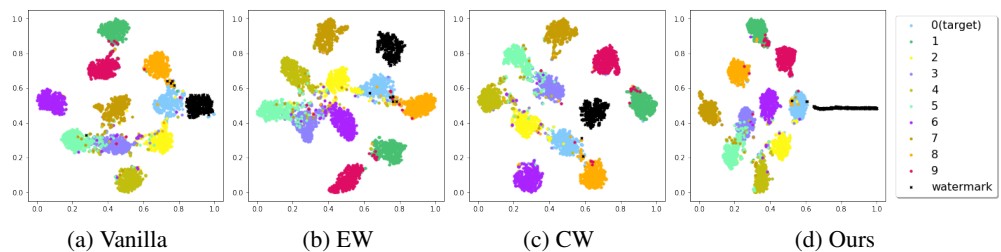

(a) Vanilla       (b) EW       (c) CW       (d) Ours

Figure 8: The t-SNE visualization of hidden feature representations.

**The Parametric Space.** We start by studying the properties of the watermarked model in the parametric space in the Introduction and illustrate how WSR changes in the vicinity of the watermarked model from the *vanilla* method. Here, we use the same visualization method to show the vicinity of the APP-based method (implementation details can be found in Appendix A. As shown in Figure 7, we find the APP-based watermarked model is able to keep WSR high within a larger range compared to the *vanilla* one (which can be seen in Figure 1). Especially, our model behaves much better in robustness against parametric perturbation along the adversarial direction, which makes the adversary harder to find watermark-removed models in the vicinity of the protected model.

**The Feature Space.** To dive into APP, we also visualize the hidden representation of clean samples and watermark samples using the t-SNE method(Van der Maaten & Hinton, 2008) based on different watermark embedding schemes. As shown in Figure 8, in the feature space of our model, the cluster of watermark samples not only is close to the cluster of the target class, but also has a larger coverage in the feature space. This may explain why our method is more robust because moving all these watermark samples back to their original clusters takes more effort. Implementation details and more results can be found in Appendix D.

## 5 CONCLUSION

In this paper, we investigated the parametric space and found there exist many watermark-removed models in the vicinity of the watermarked model, which may be easily used by removal attacks. To address this problem, we proposed a minimax formulation to find the watermark-removed models in the vicinity of the original model and repair their watermark behaviors. Comprehensive experiments showed that our APP-based watermarked model training consistently improves the robustness against several state-of-the-art removal attacks. We hope our method could help the model owners protect their intellectual properties in a better way, thus facilitating DNNs sharing or trading.

ETHICS STATEMENT

In this paper, we propose a minimax optimization-based method to embed a more robust model watermark. Our main goal is to assist the model owners to better protect their intellectual properties, which have positive social effects. However, we notice that our method may make backdoor attacks more resistant to current backdoor defenses. Accordingly, it could be used for malicious purposes. People can mitigate this threat by only using resources from reliable third parties.

REPRODUCIBILITY STATEMENT

The detailed descriptions of datasets, models, and training settings are provided in Appendix B. We provide part of the codes and some checkpoints to reproduce our main results. We will provide the remaining codes for reproducing our method upon the acceptance of the paper.

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

## A  DETAILS ABOUT VICINITY VISUALIZATION

To visualize the vicinity, we measure the watermark success rate (WSR) and benign accuracy (BA) on the panel spanned by the two directions $d_{adv}$ and $d_{FT}$. Specifically, $d_{adv}$ is the direction to erase watermark, *i.e.*, $d_{adv} = \nabla_{\boldsymbol{\theta}}\mathcal{L}(\boldsymbol{\theta}, \mathcal{D}_w)$, and $d_{FT}$ is the direction from the original watermarked model $\boldsymbol{\theta}_w$ to a fine-tuned model $\boldsymbol{\theta}_{FT}$, *i.e.*, $d_{FT} = \boldsymbol{\theta}_{FT} - \boldsymbol{\theta}_w$. We fine-tune the original model $\boldsymbol{\theta}_w$ for 40 iterations with the SGD optimizer using a learning rate 0.05 to obtain $\boldsymbol{\theta}_{FT}$. We explore the vicinity by moving the original parameter along with these two directions, recoding WSR and BA of neighbor model. For easier comparison, we use the relative distance in the parametric space, *i.e.*,

$$\boldsymbol{\theta} = \boldsymbol{\theta}_w + \alpha \frac{d_{adv}}{\|d_{adv}\|} \|\boldsymbol{\theta}_w\| + \beta \frac{d_{FT}}{\|d_{FT}\|} \|\boldsymbol{\theta}_w\|, \tag{6}$$

where $(\alpha, \beta)$ are the given coordinates. After obtaining the parameter $\boldsymbol{\theta}$ in the vicinity, we further adjust BatchNorm by re-calculating the statistic on the clean dataset to restore benign accuracy. Finally, we evaluate this neighbor model and record its benign accuracy and watermark success rate.

## B  DETAILS ABOUT MAIN EXPERIMENTS

In this section, we first briefly introduce our baseline methods, then provide the detailed settings for our main experiments. We report the full results on CIFAR-10 and CIFAR-100 at the end.

### B.1  MORE DETAILS ABOUT BASELINE METHODS

Vanilla model watermark (Zhang et al., 2018) mixed the watermark samples with the clean samples, based on which to train the model. EW (Namba & Sakuma, 2019) trained the model with exponentially reweighted parameter $EW(\theta, T)$ rather than vanilla weight $\theta$. They exponentially reweighted the $i$th element of the $l$th parameter $\theta^l$, *i.e.*,

$$EW(\theta^l, T) = \theta^l_{exp}, \text{ where } \theta^l_{exp,i} = \frac{\exp(|\theta^l_i|T)}{\max_i(\exp(|\theta^l_i|T))}\theta^l_i, \tag{7}$$

and $T$ is a hyper-parameter adjusting the intensity of the reweighting. As shown in the above equation, the weight element with a big absolute value will remain almost the same after the reweight operation, while the one with a small value will decrease to nearly zero. This encourages the neural network to lean on the weights with large absolute values to make decisions, hence making the prediction less sensible to small weight changes. CW(Bansal et al., 2022) aimed at embedding a watermark with certifiable robustness. They adopted the theory of randomized smoothing(Cohen et al., 2019) and watermarked the network using a gradient estimated with random perturbed weights. The gradient on the watermark batch $\mathcal{B}$ is calculated by

$$g_\theta = \frac{1}{k} \sum_{i=1}^{k} E_{G \in \mathcal{N}(0,(\frac{i}{k})^2 I)} E_{(x,y) \in \mathcal{B}}[\nabla l(x, y; \theta + G)], \tag{8}$$

where $\sigma$ is the noise strength.

### B.2  MORE DETAILS ABOUT WATERMARK-REMOVAL ATTACKS

**FT** (Uchida et al., 2017) removed the watermark by updating model parameters using additional holding clean data.

**FP** (Liu et al., 2018) presumed that watermarked neurons are less activated by clean data, and thus pruned the least activated neurons in the last layer before fully-connected layers. They further find-tuned the pruned model to restore benign accuracy and suppress watermarked neurons.

**ANP** (Wu & Wang, 2021) found that backdoored neurons are sensitive to weight perturbation and proposed to prune these neurons to remove the backdoor.

**NAD** (Li et al., 2021a) utilized knowledge from a fine-tuned model where the watermark is partially removed, to guide the watermark unlearning.

**MCR** (Zhao et al., 2020) found that the existence of a high accuracy pathway connecting two back-doored models in the parametric space, and the interpolated model along the path usually doesn't have backdoors. This property allows MCR to be applied in the mission of watermark removal.

**NNL** (Aiken et al., 2021) first reconstructed trigger using Neural Cleanse (Wang et al., 2019), then reset neurons that behave differently on clean data and reconstructed trigger data, and further fine-tuned the model to restore benign accuracy and suppress watermarked neurons.

### B.3    More Details about Watermark Settings

**Settings for EW.** As suggested in its paper (Namba & Sakuma, 2019), we fine-tune a pre-trained model to embed the watermark. We pre-train the model using the original dataset without injecting the watermark samples. The pre-trained model is trained for 100 epochs using the SGD optimizer with an initial learning rate of 0.1, the learning rate decays by a factor of 10 at the 50th and 75th epochs. We fine-tune the pre-trained model for 20 epochs to embed the watermark, with an initial learning rate of 0.1, and the learning rate is drop by 10 at the 10th and 15th epochs.

**Settings for CW.** For a fair comparison, we adopt a learning rate schedule and a weight-decay factor identical to other methods. Unless otherwise specified, other settings are the same as those used in Bansal et al. (2022).

**Settings for Our Method.** For the classification loss term, we calculate the loss using a batch of 128 clean samples, while for the watermark term, we use a batch of 64 clean samples and 64 watermark samples to obtain the estimation of adversarial gradients.

### B.4    More Details about Watermark-removal Settings

**Settings for FT.** We fine-tune the watermarked model for 30 epochs using the SGD optimizer with an initial learning rate of 0.05 and a momentum of 0.9, the learning rate is dropped by a factor of 0.5 every five epochs.

**Settings for FP.** We prune 90% of the least activated neurons in the last layer before fully-connected layers, and after pruning, we fine-tune the pruned model using the same training scheme as FT.

**Settings for ANP.** For ANP, we set the pruning rate to 0.6, where all defense shares a similar BA, as shown in Figure 10.

**Settings for NAD.** The original NAD only experimented on WideResNet models. In our work, we calculate the NAD loss over the output of the four main layers of ResNet, with all $\beta$s set to 1500. To obtain a better watermark removal performance, we use an initial learning rate of 0.02 , which is larger than 0.01 in the original paper (Li et al., 2021a).

**Settings for MCR.** MCR finds a backdoor-erased model on the path connecting two backdoored models. But in our settings, only one watermarked model is available. Hence the attacker must obtain the other model via fine-tuning the original watermarked model, then perform MCR using the original watermarked model and fine-tuned model. We split the attacker's dataset into two equal halves, one used to fine-tune the model and the other one to train the curve connecting the original model and the fine-tuned model. This fine-tuning is performed for 50 epochs with an initial learning rate of 0.05, which decays by a factor of 0.1 every 10 epochs. For MCR results, $t = 0$ denotes the original model and $t = 1$ denotes the original model. We select results with $t = 0.9$, where all defense shares similar BA, see Figure 9.

**Settings for NNL.** We reconstruct the trigger using Neural Cleanse (Wang et al., 2019) for 15 epochs, and reset neurons that behave significantly different under clean input and reconstructed input, we fine-tune the model for 15 epochs with the SGD optimizer, the initial learning rate is 0.02 and is divided by 10 at the 10th epoch.

### B.5    Detailed Results on CIFAR-10

The detailed results on CIFAR-10 are shown in Table 3. Moreover, we can observe from Figure 9 and Figure 10 that our method outperforms other methods regardless of the threshold value used in MCR and ANP, in terms of robustness.

Table 3: Results on CIFAR-10. 'NA' denotes 'No Attack'.

| Metric | Type | Method | NA | FT | FP | ANP | NAD | MCR | NNL | AvgDrop |
|---|---|---|---|---|---|---|---|---|---|---|
| WSR | Content | Vanilla | 99.63 | 39.91 | 66.46 | 38.78 | 23.76 | 20.38 | 13.59 | ↓ 65.81 |
| | | EW | 99.36 | 54.71 | 43.10 | 55.49 | 27.92 | 18.31 | 17.47 | ↓ 63.19 |
| | | CW | 99.70 | 20.34 | 29.83 | 0.73 | 6.10 | 18.19 | 10.00 | ↓ 85.51 |
| | | Ours | **99.92** | **98.72** | **98.64** | **99.43** | **75.87** | **78.89** | 26.12 | ↓ **20.31** |
| | Noise | Vanilla | 99.99 | 28.38 | 28.21 | 14.52 | 3.88 | 10.99 | 1.00 | ↓ 85.50 |
| | | EW | 99.99 | 5.10 | 39.35 | 28.54 | 0.04 | 0.07 | **3.34** | ↓ 87.25 |
| | | CW | 100.00 | 0.13 | 10.87 | 0.18 | 0.04 | 1.41 | 0.30 | ↓ 97.84 |
| | | Ours | 100.00 | **66.54** | **75.59** | **83.73** | **23.98** | **68.86** | 3.22 | ↓ **46.35** |
| | Unrelated | Vanilla | **100.00** | 18.82 | 24.61 | 22.31 | 2.76 | 10.91 | 67.35 | ↓ 75.54 |
| | | EW | 99.97 | 71.46 | 66.59 | 46.48 | 12.48 | 32.44 | 64.94 | ↓ 50.90 |
| | | CW | 100.00 | 9.51 | 14.17 | 3.20 | 5.28 | 5.02 | 13.41 | ↓ 91.57 |
| | | Ours | 99.95 | **96.15** | **95.46** | **99.60** | **89.28** | **87.49** | **94.49** | ↓ **6.20** |
| BA | Content | Vanilla | 93.64 | 91.84 | 92.10 | 90.63 | 90.08 | 89.24 | 91.70 | 2.71 |
| | | EW | 93.05 | 91.16 | 91.52 | 89.62 | 89.58 | 88.28 | 91.27 | 2.81 |
| | | CW | 93.46 | 91.66 | 91.70 | 87.52 | 88.66 | 88.73 | 91.32 | 3.53 |
| | | Ours | **93.79** | 91.85 | 92.14 | 88.41 | 90.35 | 89.36 | 91.15 | 3.25 |
| | Noise | Vanilla | 93.57 | 92.00 | 92.12 | 89.87 | 90.59 | 89.41 | 91.58 | 2.64 |
| | | EW | 92.99 | 91.05 | 91.41 | 89.09 | 88.81 | 88.39 | 91.14 | 3.01 |
| | | CW | **93.67** | 91.19 | 91.79 | 86.32 | 85.12 | 88.74 | 91.28 | 4.60 |
| | | Ours | 93.47 | 91.59 | 91.87 | 86.75 | 90.14 | 89.18 | 90.73 | 3.43 |
| | Unrelated | Vanilla | **93.52** | 91.53 | 91.91 | 90.16 | 89.16 | 88.22 | 90.77 | 3.23 |
| | | EW | 93.02 | 91.17 | 91.44 | 89.23 | 89.13 | 88.30 | 90.80 | 3.01 |
| | | CW | 93.47 | 91.17 | 91.29 | 86.31 | 88.97 | 87.83 | 90.72 | 4.60 |
| | | Ours | 93.30 | 91.47 | 91.46 | 86.48 | 89.70 | 89.08 | 90.36 | 3.54 |

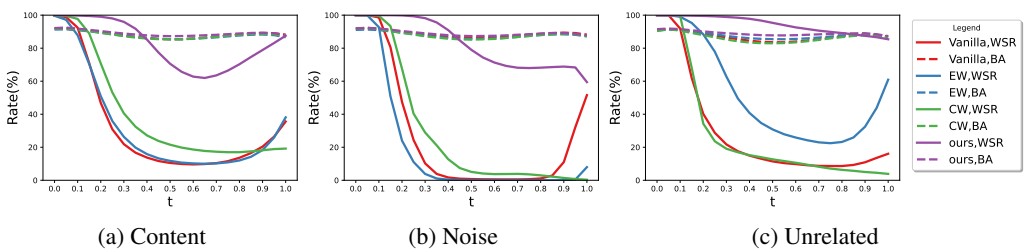

(a) Content        (b) Noise        (c) Unrelated

Figure 9: MCR results with varying thresholds on CIFAR-10.

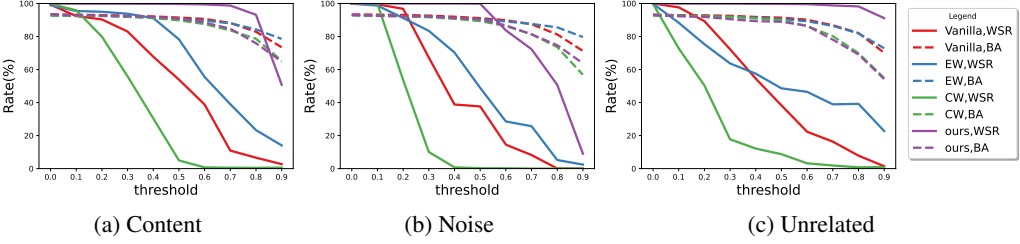

(a) Content        (b) Noise        (c) Unrelated

Figure 10: ANP results with varying thresholds on CIFAR-10.

Table 4: Results on CIFAR-100. 'NA' denotes 'No Attack'.

| Metric | Type | Method | NA | FT | FP | ANP | NAD | MCR | NNL | AvgDrop |
|--------|------|--------|----|----|----|----|----|----|----|---------|
| WSR | Content | Vanilla | 98.51 | 32.32 | 1.57 | 87.69 | 1.85 | 18.67 | 0.45 | ↓74.74 |
| | | EW | 98.21 | 18.63 | 1.44 | 88.51 | 0.79 | 2.28 | 2.52 | ↓79.18 |
| | | CW | 99.08 | 8.57 | 0.18 | 66.42 | 1.38 | 6.14 | 0.17 | ↓85.27 |
| | | Ours | **99.62** | **97.74** | **97.25** | **99.39** | **93.50** | **97.04** | **20.02** | ↓**15.46** |
| | Noise | Vanilla | 99.94 | 60.54 | **10.03** | 96.55 | 20.57 | 52.77 | 0.12 | ↓59.85 |
| | | EW | 99.87 | 10.73 | 9.79 | 95.62 | 6.69 | 8.75 | **12.99** | ↓75.78 |
| | | CW | 99.98 | 24.38 | 1.80 | 55.95 | 3.28 | 38.44 | 0.05 | ↓79.33 |
| | | Ours | **100.00** | **84.82** | 8.60 | **99.99** | **73.67** | **93.82** | 0.98 | ↓**39.69** |
| | Unrelated | Vanilla | **100.00** | 6.83 | 1.50 | 92.25 | 6.25 | 12.58 | 11.42 | ↓78.19 |
| | | EW | **100.00** | 27.67 | 3.42 | 93.33 | 18.25 | 17.75 | 40.25 | ↓66.56 |
| | | CW | 99.83 | 0.25 | 1.08 | 41.08 | 4.08 | 7.67 | 0.58 | ↓90.71 |
| | | Ours | **100.00** | **97.42** | **44.67** | **100.00** | **94.08** | **97.25** | **45.17** | ↓**20.24** |
| ACC | Content | Vanilla | 74.09 | 69.51 | 68.57 | 71.00 | 65.89 | 64.05 | 67.58 | 6.33 |
| | | EW | 73.75 | 67.49 | 66.97 | 69.29 | 63.91 | 60.98 | 65.69 | 8.03 |
| | | CW | 73.75 | 68.14 | 67.74 | 51.27 | 63.06 | 61.99 | 66.74 | 10.59 |
| | | Ours | 73.69 | 68.80 | 68.19 | 69.07 | 65.68 | 63.75 | 67.13 | 6.59 |
| | Noise | Vanilla | 74.13 | 69.61 | 68.78 | 70.72 | 66.30 | 63.73 | 67.30 | 6.39 |
| | | EW | 73.43 | 67.39 | 66.92 | 68.85 | 64.18 | 61.10 | 66.96 | 7.53 |
| | | CW | 73.49 | 68.00 | 67.84 | 59.21 | 64.26 | 61.68 | 66.79 | 8.86 |
| | | Ours | 72.97 | 68.49 | 67.39 | 67.59 | 64.94 | 63.08 | 66.25 | 6.68 |
| | Unrelated | Vanilla | 73.80 | 68.55 | 67.46 | 69.90 | 65.14 | 61.87 | 65.77 | 7.35 |
| | | EW | 73.57 | 67.83 | 66.61 | 69.39 | 63.52 | 61.47 | 65.90 | 7.78 |
| | | CW | 73.45 | 67.45 | 66.90 | 54.59 | 62.66 | 60.60 | 64.88 | 10.60 |
| | | Ours | 72.27 | 67.68 | 66.88 | 65.22 | 64.07 | 61.99 | 62.64 | 7.53 |

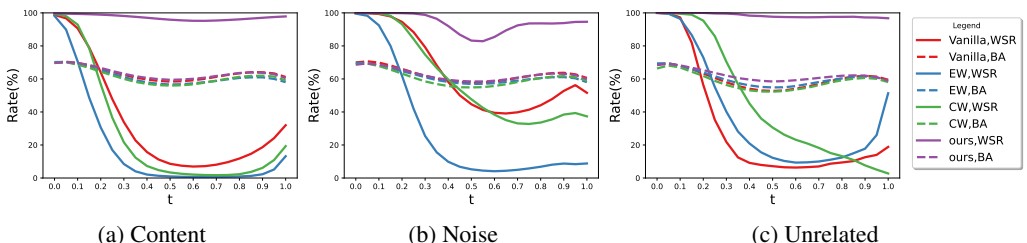

(a) Content      (b) Noise      (c) Unrelated

Figure 11: MCR results with varying thresholds on CIFAR-100.

## B.6 DETAILED RESULTS ON CIFAR-100

To verify that our model can apply to other datasets, we experiment on CIFAR-100, and the results are shown as follows.

**Modification to Attack Settings.** As trigger reconstruction need to scan 100 classes on CIFAR-100, we reduce the NC reconstruction epoch from 15 to 5 to speed it up. The ANP pruning threshold is set to 0.5 in CIFAR-100 experiments to maintain benign accuracy.

**Results.** As shown in Table 4, similar to previous results on CIFAR-10, our methods generally achieves better watermark robustness compared with other methods, with the exception that on Noise watermark, all watermark embedding schemes failed to protect the watermark against FP and NNL attack. Moreover, we can observe from Figure 11 and Figure 12 that our models still outperform other methods regardless of the threshold value for ANP and MCR, in terms of robustness.

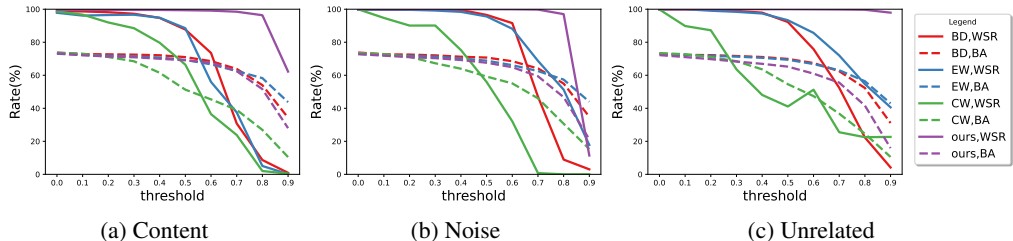

|  | (a) Content | (b) Noise | (c) Unrelated |

Figure 12: ANP results with varying thresholds on CIFAR-100.

Table 5: Results of Content embedded with varying perturbation magnitude $\epsilon$ using our method. AVG denotes the average WSR/BA after watermark removal attacks.

| Metric | $\epsilon$ | NA | FT | FP | ANP | NAD | MCR | NNL | AVG |
|--------|-----------|------|------|------|------|------|------|------|------|
| WSR | $5 \times 10^{-3}$ | 99.88 | 92.84 | 96.28 | 96.97 | 37.56 | 39.11 | 26.68 | 57.32 |
|  | $1 \times 10^{-2}$ | 99.84 | 95.76 | 97.26 | 97.79 | 75.36 | 49.16 | 38.69 | 67.35 |
|  | $2 \times 10^{-2}$ | 99.92 | **98.72** | 98.64 | 99.43 | 75.87 | 78.89 | 26.12 | 70.38 |
|  | $4 \times 10^{-2}$ | 99.93 | 98.71 | **99.12** | **99.76** | **89.21** | **96.44** | **69.71** | **88.39** |
|  | $8 \times 10^{-2}$ | 99.76 | 84.24 | 80.51 | 97.40 | 53.90 | 67.18 | 18.17 | 67.58 |
| BA | $5 \times 10^{-3}$ | 93.81 | 91.83 | 92.15 | 88.10 | 90.19 | 89.42 | 91.84 | 90.71 |
|  | $1 \times 10^{-2}$ | 93.73 | 91.80 | 92.03 | 88.92 | 89.94 | 89.27 | 91.39 | 90.80 |
|  | $2 \times 10^{-2}$ | 93.79 | 91.85 | 92.14 | 88.41 | 90.35 | 89.36 | 91.15 | 90.84 |
|  | $4 \times 10^{-2}$ | 93.36 | 91.56 | 91.79 | 87.27 | 90.03 | 89.29 | 90.83 | 90.34 |
|  | $8 \times 10^{-2}$ | 93.14 | 91.35 | 91.48 | 86.91 | 89.59 | 89.04 | 90.24 | 90.14 |

## C  Detailed Results of Ablation Studies

### C.1  Results with Varying Perturbation Magnitude

We visualize some results of the Content watermark embedded with different perturbation magnitude $\epsilon$ in Sec 4.3. Here, we provided more detailed results in a numeric form in Table 5. Generally speaking, our method consistently improves the robustness of the watermark, with the watermark success rate higher than other methods throughout all tested $\epsilon$. Moreover, the amount of improvement against all evaluated attacks shows similar trends, and this consistent robustness improvement benefits the selection of perturbation magnitude $\epsilon$. We also notice that the most robust watermark is obtained with $\epsilon = 4 \times 10^{-2}$, rather than the default setting $\epsilon = 2 \times 10^{-2}$, indicating that a good $\epsilon$ especially selected for the chosen watermark type may further improve the robustness.

### C.2  Results with Other Target Classes

To demonstrate that our method can apply to different target classes, we experimented with Content and set the target class $y_t \in \{1, 2, 3, 4\}$. Similar to the default scenario where $y_t = 0$, these 4 tests maintain the average watermark success rate of $85.69\%$, $72.99\%$, $85.72\%$, and $82.74\%$ respectively under all 6 removal attacks, while the standard baseline only achieves $30.18\%$, $10.90\%$, $30.16\%$, and $18.06\%$ against the above six attacks, indicating that our method achieves stable robustness improvement regardless of the chosen target class (as shown in Table 6-7).

## D  Visualizing the Feature Space

To provide further understandings about the effectiveness of our method, we visualize the how the hidden representation evolves along the adversarial direction and during the process of fine-tuning via t-SNE(Van der Maaten & Hinton, 2008).

Table 6: Results of standard model watermark over content-type attack with different target labels.

| Metric | $y_t$ | NA | FT | FP | ANP | NAD | MCR | NNL | AVG |
|--------|-------|-----|-----|-----|-----|-----|-----|-----|-----|
| WSR | 0 | 99.63 | 39.91 | 66.46 | 38.78 | 23.76 | 20.38 | 13.59 | 33.81 |
| | 1 | 99.39 | 40.81 | 57.73 | 35.18 | 16.33 | 23.16 | 7.85 | 30.18 |
| | 2 | 99.52 | 15.77 | 37.81 | 3.36 | 2.03 | 4.47 | 1.97 | 10.90 |
| | 3 | 99.44 | 73.24 | 71.71 | 6.35 | 14.29 | 9.54 | 5.85 | 30.16 |
| | 4 | 99.40 | 34.39 | 38.97 | 11.81 | 7.66 | 10.72 | 4.81 | 18.06 |
| BA | 0 | 93.64 | 91.84 | 92.10 | 90.63 | 90.08 | 89.24 | 91.70 | 90.93 |
| | 1 | 93.61 | 91.94 | 92.05 | 89.85 | 90.45 | 89.34 | 91.79 | 90.90 |
| | 2 | 93.82 | 91.99 | 92.15 | 90.09 | 90.59 | 89.06 | 91.97 | 90.98 |
| | 3 | 93.83 | 92.06 | 92.19 | 90.01 | 90.42 | 89.19 | 91.66 | 90.92 |
| | 4 | 93.72 | 92.04 | 92.30 | 89.92 | 90.55 | 89.26 | 91.52 | 90.93 |

Table 7: Results of our model watermark over content-type attack with different target labels.

| Metric | $y_t$ | NA | FT | FP | ANP | NAD | MCR | NNL | AVG |
|--------|-------|-----|-----|-----|-----|-----|-----|-----|-----|
| WSR | 0 | 99.92 | 98.72 | 98.64 | 99.43 | 75.87 | 78.89 | 26.12 | 82.51 |
| | 1 | 99.84 | 97.84 | 98.10 | 99.37 | 91.17 | 71.56 | 41.96 | 85.69 |
| | 2 | 99.84 | 97.79 | 98.27 | 99.38 | 40.47 | 72.34 | 2.85 | 72.99 |
| | 3 | 99.79 | 99.05 | 98.64 | 99.00 | 85.96 | 79.61 | 38.00 | 85.72 |
| | 4 | 99.79 | 98.24 | 98.30 | 99.30 | 87.35 | 77.21 | 19.01 | 82.74 |
| BA | 0 | 93.79 | 91.85 | 92.14 | 88.41 | 90.35 | 89.36 | 91.15 | 91.01 |
| | 1 | 93.57 | 91.82 | 91.87 | 88.64 | 90.22 | 88.89 | 91.37 | 90.91 |
| | 2 | 93.59 | 91.68 | 91.92 | 89.02 | 90.11 | 89.21 | 90.92 | 90.92 |
| | 3 | 93.46 | 91.70 | 91.86 | 87.49 | 90.18 | 89.24 | 91.31 | 90.75 |
| | 4 | 93.51 | 91.68 | 91.77 | 88.80 | 90.00 | 88.92 | 91.06 | 90.82 |

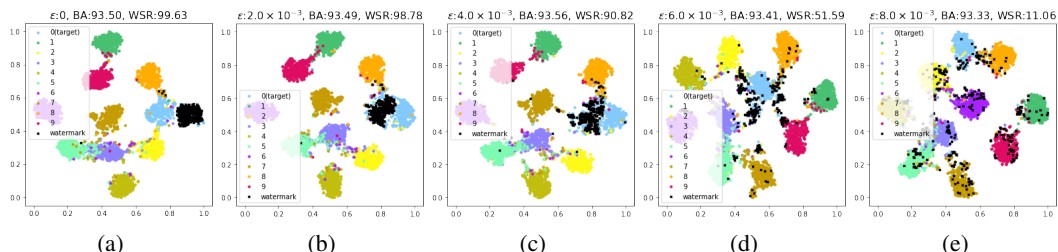

| (a) | (b) | (c) | (d) | (e) |

Figure 13: t-SNE visualization of vanilla watermarked model along the adversarial direction.

### D.1 FEATURES ALONG WITH THE ADVERSARIAL DIRECTION

To show how the hidden representation evolves along the adversarial direction, we add a small adversarial perturbation to the watermarked model with the perturbation magnitude growing by $2 \times 10^{-3}$ every step. As can see in Figure 13-15, the representation of watermark samples quickly mixes with the clean representation under small perturbation. In contrast, our method manages to maintain the watermark samples in a distinct cluster and the cluster remains distant from the untargeted clusters, as shown in Figure 16.

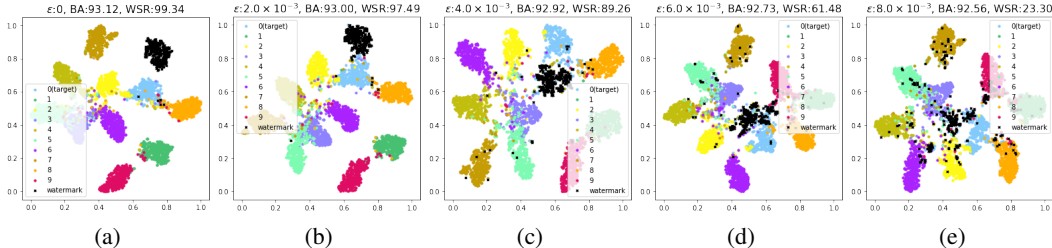

Figure 14: t-SNE visualization of EW watermarked model along the adversarial direction.

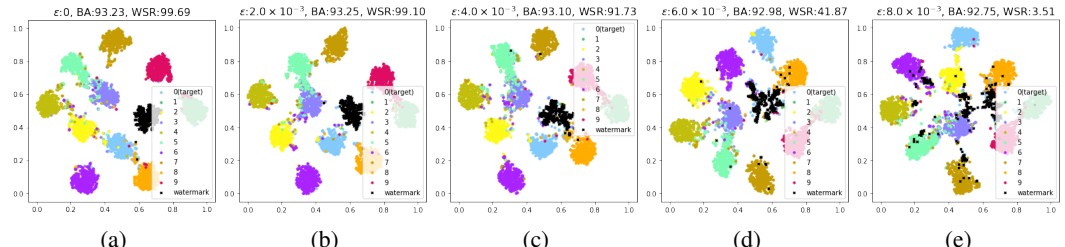

Figure 15: t-SNE visualization of CW watermarked model along the adversarial direction.

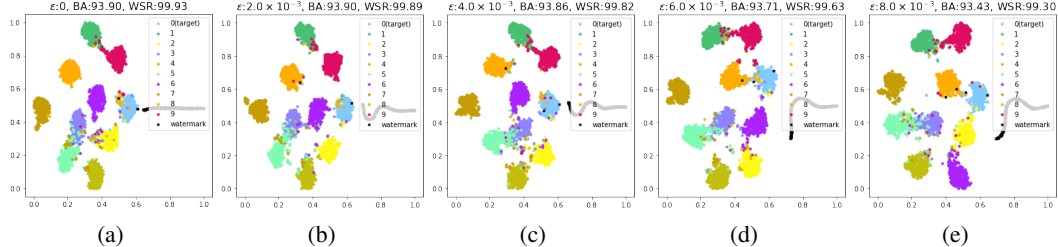

Figure 16: t-SNE visualization of our watermarked model along the adversarial direction.

### D.2 FEATURE EVOLUTION DURING THE PROCESS OF FINE-TUNING

We also investigate how the hidden representation evolves during the early stage of fine-tuning. We fine-tune the watermarked models for 200 iterations using the SGD optimizer with a learning rate of 0.05 and show how the representation evolves via t-SNE every 50 iterations. As can see in Figure 17-19, the representation of watermark samples quickly mixes with the clean representation in the early phase of fine-tuning, with the watermark success rate decreasing. While our method manages to maintain the watermark samples in a distinct cluster, and the cluster stays distant from the untargeted clusters during the fine-tuning process, as shown in Figure 20.

## E ADDITIONAL RESULTS OF OTHER BASELINE DEFENSES

In our main experiments, we only compared our method with two SOTA methods (*i.e.*, Namba & Sakuma (2019) and Bansal et al. (2022)), out of four methods in total mentioned in Section 2. These two compared methods and ours have a similar threat model. In this section, we provide additional results for the two remaining baseline defenses.

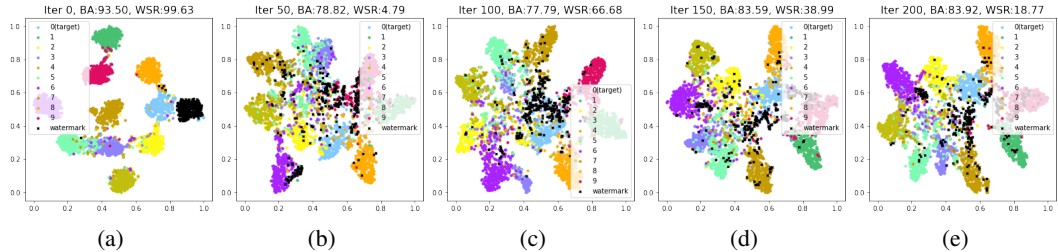

Figure 17: t-SNE visualization of vanilla watermarked model during the process of fine-tuning.

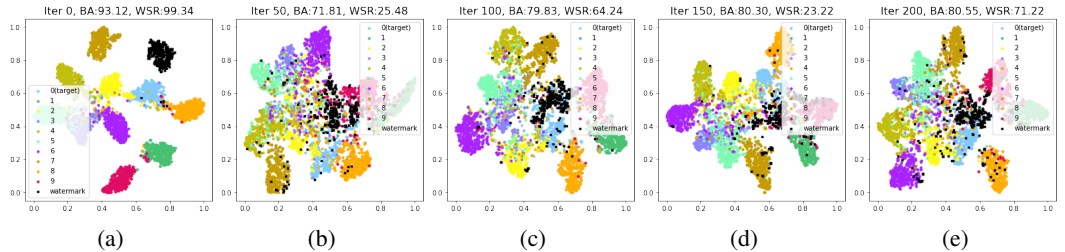

Figure 18: t-SNE visualization of EW watermarked model during the process of fine-tuning.

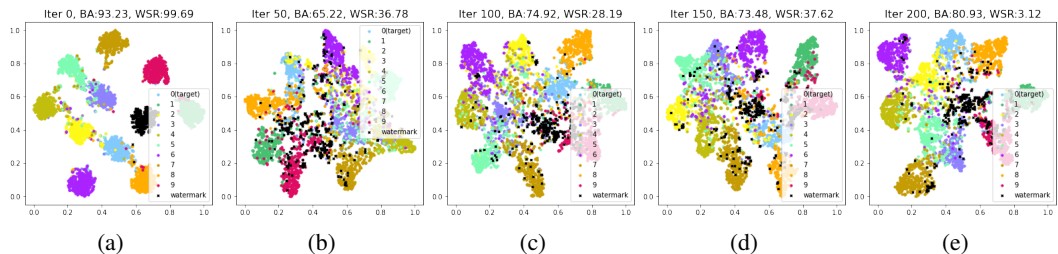

Figure 19: t-SNE visualization of CW watermarked model during the process of fine-tuning.

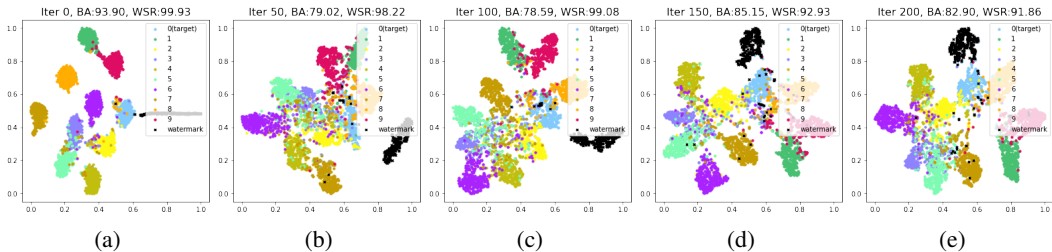

Figure 20: t-SNE visualization of our watermarked model during the process of fine-tuning.

### E.1 PIRACY RESISTANT WATERMARK FOR DEEP NEURAL NETWORKS

Li et al. (2019) verifies its ownership by querying the suspect model with extremely large pixel values (*i.e.*, 2,000) that far exceed the range of normal pixel values (*i.e.*, [0,1]). This defense can be easily circumvented by clipping image pixels to the normal range (*i.e.*, [0,1]) or refusing the predictions of these abnormal samples. This defense is only feasible for models without normalization layers (*e.g.*, batch normalization (Ioffe & Szegedy, 2015)). As shown in Table 8, training VGG-16 with batch normalization using this method will lead to very low benign accuracy on CIFAR-10.

Table 8: The results of (Li et al., 2019) with different model architectures.

| Model↓, Metric→ | BA | WSR |
|---|---|---|
| VGG-16 (w/o BN) | 89.35 | 100.00 |
| VGG-16 (w/ BN) | 10.00 | 100.00 |

Table 9: The results of CAE on CIFAR-10.

| Metric | Method | NA | FT | FP | ANP | NAD | MCR | NNL | AvgDrop |
|---|---|---|---|---|---|---|---|---|---|
| WSR | *Vanilla* | 99.63 | 39.91 | 66.46 | 38.78 | 23.76 | 20.38 | 13.59 | ↓65.81 |
| | CAE | 100.00 | 67.67 | 67.67 | 61.33 | **78.67** | 68.33 | 23.33 | ↓38.83 |
| | Ours | **99.92** | **98.72** | **98.64** | **99.43** | 75.87 | **78.89** | **26.12** | ↓**20.31** |
| BA | *Vanilla* | 93.64 | 91.84 | 92.10 | 90.63 | 90.08 | 89.24 | 91.70 | 2.71 |
| | CAE | **93.80** | 91.97 | 92.32 | 90.20 | 90.47 | 89.65 | 91.49 | 2.78 |
| | Ours | 93.79 | 91.85 | 92.14 | 88.41 | 90.35 | 89.36 | 91.15 | 3.25 |

We conjuncture that this failure is mostly because the batch statistics (mean & variance) calculated inside batch normalization are unduly affected by outliers caused by these extreme values in input pixels. We will further explore this problem in our future work.

### E.2 DEEP NEURAL NETWORK FINGERPRINTING BY CONFERRABLE ADVERSARIAL EXAMPLES

Different from existing methods that adopted predefined patterns to generate watermark samples, Lukas et al. (2020) exploited conferrable adversarial examples (CAE) as watermark samples. Specifically, it was an ensemble-based defense requiring training 36 different models. Accordingly, is very time-consuming, requiring a large amount of training resources.

In this section, we compare CAE with our method using content-type watermark samples. As shown in Table 9, our method is still (significantly) better than CAE in most cases (5 out of 6). These results verify the effectiveness of our method again.

## F ADDITIONAL RESULTS OF OTHER POTENTIAL BASELINES

Recall that our method exploits a min-max formulation with respect to model parameters. One may wonder whether it would be better to use random parametric perturbations instead of adversarial ones or use standard adversarial training in the input space. In this section, we use the content-based watermark on CIFAR-10 as an example for our discussions.

### F.1 USING RANDOM INSTEAD OF ADVERSARIAL PARAMETRIC PERTURBATION

To explore whether using random parametric perturbation (RPP) is better than our defense, we use a random parametric perturbation instead of the adversarial parametric perturbation within the minimization $w.r.t.$ model parameter $\theta$. As shown in Table 10, although RPP achieves some improvements over the model trained without any defense (*i.e.*, Vanilla), our method is still significantly better than it in almost all cases. These results verify the effectiveness of our method again.

### F.2 USING STANDARD ADVERSARIAL TRAINING IN THE INPUT SPACE

To explore whether traditional adversarial training (AT) is better than our method, we conduct additional experiments by performing traditional AT in the input space. In particular, we adopt AT on the watermark instead of all samples to preserve high benign accuracy. As shown in Table 10, our method is still significantly better than AT, although it has mild improvement compared to training with no defense (*i.e.*, Vanilla) in some cases.

Table 10: The results of RPP and AT on CIFAR-10.

| Metric | Method | NA | FT | FP | ANP | NAD | MCR | NNL | AvgDrop |
|--------|--------|-----|-----|-----|------|------|------|------|---------|
| WSR | *Vanilla* | 99.63 | 39.91 | 66.46 | 38.78 | 23.76 | 20.38 | 13.59 | ↓ 65.81 |
| | RPP | 99.78 | 68.19 | 70.57 | 81.76 | 19.03 | 12.87 | 7.76 | ↓ 56.42 |
| | AT | 99.93 | 74.84 | 73.85 | 33.22 | 24.05 | 27.77 | 8.94 | ↓ 59.48 |
| | Ours | **99.92** | **98.72** | **98.64** | **99.43** | **75.87** | **78.89** | **26.12** | **↓ 20.31** |
| BA | *Vanilla* | 93.64 | 91.84 | 92.10 | 90.63 | 90.08 | 89.24 | 91.70 | 2.71 |
| | RPP | 93.87 | 92.02 | 92.38 | 89.91 | 90.34 | 89.40 | 91.85 | 2.89 |
| | AT | 93.69 | 91.98 | 92.06 | 89.83 | 90.42 | 89.14 | 91.54 | 2.86 |
| | Ours | **93.79** | 91.85 | 92.14 | 88.41 | 90.35 | 89.36 | 91.15 | 3.25 |

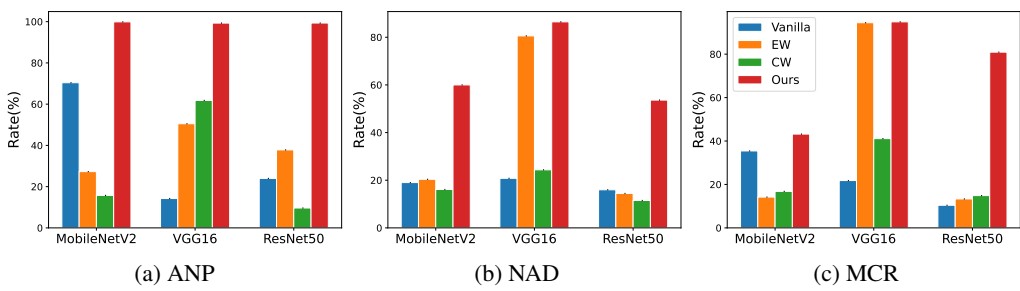

(a) ANP      (b) NAD      (c) MCR

Figure 21: The WSR of models under ANP, NAD, and MCR.

Table 11: The results with MobileNetV2 on CIFAR-10.

| Metric | Method | NA | FT | FP | ANP | NAD | MCR | NNL | AvgDrop |
|--------|--------|-----|-----|-----|------|------|------|------|---------|
| WSR | *Vanilla* | 99.19 | 11.58 | 21.82 | 70.39 | 19.01 | 35.43 | 7.10 | ↓ 71.63 |
| | EW | 98.92 | 14.14 | 17.19 | 27.28 | 20.38 | 14.18 | 10.60 | ↓ 81.62 |
| | CW | 99.06 | 14.96 | 22.03 | 15.74 | 16.11 | 16.80 | 6.71 | ↓ 83.67 |
| | Ours | **99.83** | **42.08** | **47.64** | **99.90** | **59.99** | **43.19** | **16.89** | **↓ 48.21** |
| BA | *Vanilla* | **92.29** | 89.45 | 90.09 | 60.51 | 87.73 | 85.29 | 89.26 | 8.57 |
| | EW | 89.90 | 86.61 | 87.32 | 79.83 | 83.36 | 80.13 | 87.38 | 5.79 |
| | CW | 92.20 | 88.98 | 89.45 | 59.36 | 86.98 | 84.06 | 88.94 | 9.24 |
| | Ours | 90.78 | 87.98 | 87.81 | 55.12 | 85.97 | 82.66 | 87.49 | 9.60 |

# G    ADDITIONAL RESULTS ON OTHER MODEL ARCHITECTURES

In Section 4.3, we demonstrate that our method improves watermark robustness against the FT attack across various model architectures (*i.e.*, MobileNetV2, VGG16, and ResNet50). To further verify that our method is better than baseline defenses across different model architectures under different attacks, in this section, we conduct additional experiments under more attacks (*i.e.*, ANP, NAD, MCR) other than FT-based attacks. As shown in Figure 21, our method consistently improves the watermark robustness across different model architectures under all attacks.

In addition, to further verify that our method is still effective under simpler model architecture, we conduct additional experiments on CIFAR-10 with MobileNetV2. MobileNetV2 consists of 2.2M trainable parameters, which is significantly less than the 11.2M parameters contained in ResNet18 used in our main experiments. As shown in Table 11, in this case, our method is still better than all baseline methods with the average WSR drop of 48.21%, whereas all baseline defenses suffer from at least 71.63% average WSR decreases. These results verify the effectiveness of our method again.

