# OpenReview forum: "Towards Robust Model Watermark via Reducing Parametric Vulnerability"
_ICLR.cc/2023/Conference — Submitted to ICLR 2023_

### Official Review · Reviewer_y4Xt · 2022-10-25

**Confidence:** 4
**Clarity, Quality, Novelty And Reproducibility:** It is understandable to a large exten…
**Correctness:** 2
**Technical Novelty And Significance:** 2
**Empirical Novelty And Significance:** 2
**Recommendation:** 3

**Strength And Weaknesses:**

Pros:
1.	This paper validated the vulnerability in the watermark of deep neural networks and shows that it is necessary to develop a robust watermark framework.
2.	This paper solves the inconsistency issue during the training by introducing a clean-sample-based BatchNorm.
3.	Extensive experiments show that the proposed method achieves superior robustness of watermark verification.

Cons:
1.	I have concerns about the approximation of perturbation for parameters. Equation 3 aims to find the worst case of the delta that can maximize the loss over parameters.t I do not see the clues that equation 5 is an approximation form of equation 3. It looks like equation 5 randomly selects a point (depending on the epsilon) within the vicinity of the original model.
2.	Only minimizing the worst case is not theoretically enough. The author should select multiple neighbor points in parameter space for each training iteration.
3.	The ablation study is not enough. Figure 6 only shows results against FT attacks.
4.	The paper controls the perturbation size with epsilon parameters. It is hard to ensure that the l norm of perturbation is within a fixed bound cause theta in equation 5 constantly changes during training.
5.	It would be good to show the results on more complex datasets like ImageNet or CIFAR10 with a simpler architecture. Because we all know that a slight change of parameters can significantly impact output, the proposed method work on CIFAR10 might be because the res-net is too redundant for CIFAR10 such that small changes in parameters have no impact on it.
6.	It is unclear for how to distinguish the proposed method from traditional adversarial training method. The SOTA adversarial training methods can still be applied in the context of the watermark of deep neural networks.

**Summary Of The Paper:**

This paper proposed an adversarial training-based framework to ensure robustness in the watermark of deep neural networks. Specifically, a normalized gradient method is applied to find the worst case within the vicinity of the original model such that we can minimize the loss in terms of noisy parameters. A clean sample-based BatchNorm is also proposed to improve consistency during the training process.

**Summary Of The Review:**

In general, the studied problem is interesting and important. In addition, the methodology is principled with three major merits as discussed above. However, the work still has some unaddressed concerns to well justify its technical and empirical contributions.

---

> ### Author Response · Authors · 2022-11-13
> **Author Response (Part I)**
>
>
> We sincerely thank you for your valuable time and comments. We are encouraged by your positive comments on our **research significance**, **extensive experiments**, **method effectiveness**! We are deeply sorry for the misunderstandings that our paper may cause you. Please kindly find our clarifications below to your concerns.
>
>
> ---
> **Q1**: I have concerns about the approximation of perturbation for parameters. Equation 3 aims to find the worst case of the delta that can maximize the loss over parameters. I do not see the clues that equation 5 is an approximation form of equation 3. It looks like equation 5 randomly selects a point (depending on the epsilon) within the vicinity of the original model.
>
>
> **R1**：In Equation (5), to approximate the worst case of perturbation $\delta$, we use a first-order approximation and let the perturbation along the gradient vector. This is **the direction of the steepest ascent rather than a random direction**. Specifically,
>
> $${\delta} = \epsilon \Vert {\theta} \Vert \cdot \frac{\nabla_\theta \mathcal{L}(\theta, \mathcal{D}_w)}{\Vert \nabla_\theta \mathcal{L}(\theta, \mathcal{D}_w) \Vert},$$
>
> where $\frac{\nabla_\theta \mathcal{L}(\theta, \mathcal{D}_w)}{\Vert \nabla_\theta \mathcal{L}(\theta, \mathcal{D}_w) \Vert}$ is the normalized direction vector whose length equals 1, and $\epsilon \Vert \theta \Vert$ controls the magnitute of the perturbation in a relative way (the reason for relative magnitude instead of absolute magnitude can be found in Q4&R4). In the reversion, we re-arrange the order of items in Equation (5) and add some descriptions to make it more clear.
>
>
>
> ---
> **Q2**: Only minimizing the worst case is not theoretically enough. The author should select multiple neighbor points in parameter space for each training iteration.
>
>
> **R2**: Thank you for your question and we do understand your concerns. We will alleviate your concerns from two main aspects, as follows:
>
> - **Theoretically, optimizing on worst-case parametric perturbation can improve watermark robustness against random weight perturbations.** Specifically, let $\mathcal{N}$ containing some random neighbor points within the neighboorhood $\mathcal{B}$, _i.e._, $\mathcal{N} \subset \mathcal{B}$. We can easily prove that
>
>
> $\max_{\theta \in \mathcal{B} } \mathcal{L}(\theta, \mathcal{D_w}) \geq \frac{1}{\vert \mathcal{N} \vert} \cdot \sum_{\theta \in \mathcal{N}} \mathcal{L}(\theta, \mathcal{D}_w).$
>
>
>
> In other words, minimizing the left-side (_i.e._, our method) itself can also minimize the right side (to some extent).
> - **Empirically, our method is significantly better than using random parametric perturbations (RPP)**. Specifically, we conduct experiments on CIFAR-10 using the aforementioned average-based instead of the maximization-based equation within the minimization $w.r.t.$ model parameter $\theta$. As shown in the following table, while RPP has mild improvement compared to training with no defense ($i.e.$, Vanilla) in some cases, our method (the worst-case miminization) performs significantly better than RPP.
>
>
> | Method| BA    | WSR   | FT    | FP    | ANP   | NAD   | MCR   | NNL   | AvgDrop  |
> |-----|-------|-------|-------|-------|-------|-------|-------|-------|----------|
> |  Vanilla  | 93.64 | 99.63 | 39.91 | 66.46 | 38.78 | 23.76 | 20.38 | 13.59 |  $\downarrow$ 65.81    |
> |  RPP  | **93.87** | 99.78 | 68.19 | 70.57 | 81.75 | 19.03 | 12.87 | 7.76 |  $\downarrow$ 56.42    |
> |  Ours  | 93.79 | **99.92** | **98.72** | **98.64** | **99.43** | **75.87** | **78.89** | **26.12** |  $\downarrow$ **20.31**    |
>
> ---

---

> > ### Author Response · Authors · 2022-11-13
> > **Author Response (Part II)**
> >
> > **Q3**: Only using FT is insufficient to demonstrate the improvement of robustness aross various model architectures, ablation study with more attacks is required.
> >
> > **R3**: To verify the effectiveness of our method across various model architectures under different attacks, we conduct additional experiments under more attacks ($i.e.$, ANP, NAD, MCR) other than FT-based attacks. The results can be found in the following tables. We find that our method consistently performs better than other baselines across various architectures (from compact model to large model) and different SOTA removal attacks.
> >
> >
> > Table 1. The WSR of models under ANP.
> > |         | MobileNetV2 | VGG16 | ResNet50  |
> > |---------|-------------|-------|-----------|
> > | Vanilla | 70.39       | 14.23 | 23.99     |
> > | EW      | 27.28       | 50.55 | 37.81     |
> > | CW      | 15.74       | 61.80 | 9.63      |
> > | Ours    | **99.90**   | **99.30** | **99.33**     |
> >
> > Table 2. The WSR of models under NAD.
> > |         | MobileNetV2 | VGG16 | ResNet50  |
> > |---------|-------------|-------|-----------|
> > | Vanilla | 19.01       | 20.80 | 15.98     |
> > | EW      | 20.38       | 80.57 | 14.44     |
> > | CW      | 16.11       | 24.36 | 11.50     |
> > | Ours    | **59.99**   | **86.46** | **53.63**     |
> >
> > Table 3. The WSR of models under MCR.
> > |         | MobileNetV2 | VGG16 | ResNet50  |
> > |---------|-------------|-------|-----------|
> > | Vanilla | 35.43       | 21.79 | 10.41     |
> > | EW      | 14.18       | 94.47 | 13.33     |
> > | CW      | 16.80       | 41.10 | 14.86     |
> > | Ours    | **43.19**       | **94.87** | **80.92**     |
> >
> >
> > ---
> > **Q4**: The paper controls the perturbation size with epsilon parameters. It is hard to ensure that the l norm of perturbation is within a fixed bound cause theta in equation 5 constantly changes during training.
> >
> > **R4**: We admit the perturbation budget may change during training. However, we restrict the parametric perturbation $\delta$ using a relative size (_e.g._ $\Vert \delta \Vert \leq \epsilon \Vert \theta \Vert$) instead of a fixed bound for two main reasons:
> > 1. Different from the common case of adversarial training where the input image is always fixed across different architectures and bounded in [0, 1], **the norm of model parameters may vary across different architectures**, necessitating case-by-case tuning of the perturbation bound for each architecture if a fixed bound is adopted. We hope our method can be easily extended to different situations without a heavy workload on hyper-parameter tuning. Thus, we choose a relative bound strategy.
> > 2. Modern architectures, especially those with NormLayers, are scale-invariant [1, 2]. For example, if we multiply every weight parameter of ResNet by a large value ($e.g.$, 10)  during training, BatchNorm will cancel out this scaling by using larger statistics to normalize the intermediate representations and make the model have the same outputs without scaling. This also motivates us to **apply a relative perturbation budget to alleviate the effects from scale invariance**.
> >
> >
> > [1] Laurent Dinh, Razvan Pascanu, Samy Bengio, Yoshua Bengio. Sharp Minima Can Generalize For Deep Nets. In ICML, 2017.
> >
> >
> > [2] Hao Li, Zheng Xu, Gavin Taylor, Christoph Studer, Tom Goldstein. Visualizing the Loss Landscape of Neural Nets. In NeurIPS, 2018.
> >
> > ---

---

> > > ### Author Response · Authors · 2022-11-13
> > > **Author Response (Part III)**
> > >
> > > **Q5**: It would be good to show the results on more complex datasets or CIFAR10 with a simpler architecture. Because we all know that a slight change of parameters can significantly impact output, the proposed method work on CIFAR10 might be because the ResNet is too redundant for CIFAR10 such that small changes in parameters have no impact on it.
> > >
> > > **R5**: Thank you for this constructive suggestion! To verify the general effectiveness of our method rather than a special case (a redundant model on a simple dataset), we conduct two additional experiments as follows:
> > >
> > > - **Results with a simpler architecture.** We evaluate our method on CIFAR-10 with a compact architecture ($i.e.$, MobileNetV2), containing 2.2M trainable parameters ($\ll$ 11.2M in ResNet18). As shown in the following table, our method achieves better performance than all other baselines, which indicates our method can generalize well to compact models.
> > >
> > > | Method  | BA    | WSR   | FT    | FP    | ANP   | NAD   | MCR   | NNL   | AvgDrop  |
> > > |---------|-------|-------|-------|-------|-------|-------|-------|-------|----------|
> > > | Vanilla |**92.29** | 99.19 | 11.58 | 21.82 | 70.39 | 19.01 | 35.43 | 7.10  | $\downarrow$ 71.63    |
> > > | EW      | 89.90 | 98.92 | 14.14 | 17.19 | 27.28 | 20.38 | 14.18 | 10.60 | $\downarrow$ 81.62    |
> > > | CW      | 92.20 | 99.06 | 14.96 | 22.03 | 15.74 | 16.11 | 16.80 | 6.71  | $\downarrow$ 83.67    |
> > > | Ours    | 90.78 | **99.83** | **42.08** | **47.64** | **99.90** | **59.99** | **43.19** | **16.89** | $\downarrow$ **48.21**    |
> > >
> > > - **Results on a more complex dataset.** We conduct additional experiments with ResNet18 on CIFAR-100, which is more complex with more classes compared to CIFAR-10. As shown in the following table, our method is also better than other baselines on the complex dataset.
> > >
> > >
> > > | Type    | Method | BA    | WSR    | FT    | FP    | ANP    | NAD   | MCR   | NNL   | AvgDrop  |
> > > |---------|--------|-------|--------|-------|-------|--------|-------|-------|-------|----------|
> > > | Content | BD     | **74.09** | 98.51  | 32.32 | 1.57  | 87.69  | 1.85  | 18.67 | 0.45  | $\downarrow$ 74.75    |
> > > |         | EW     | 73.75 | 98.21  | 18.63 | 1.44  | 88.51  | 0.79  | 2.28  | 2.52  | $\downarrow$ 79.18    |
> > > |         | CW     | 73.75 | 99.08  | 8.57  | 0.18  | 66.42  | 1.38  | 6.14  | 0.17  | $\downarrow$ 85.27    |
> > > |         | Ours   | 73.69 | **99.62**  | **97.74** | **97.25** | **99.39**  | **93.50** | **97.04** | **20.02** | $\downarrow$ **15.46**    |
> > > | Noise   | BD     | **74.13** | 99.94  | 60.54 | **10.03** | 96.55  | 20.57 | 52.77 | 0.12  | $\downarrow$ 59.85    |
> > > |         | EW     | 73.43 | 99.87  | 10.73 | 9.79  | 95.62  | 6.69  | 8.75  | **12.99** | $\downarrow$ 75.78    |
> > > |         | CW     | 73.49 | 99.98  | 24.38 | 1.80  | 55.95  | 3.28  | 38.44 | 0.05  | $\downarrow$ 79.33    |
> > > |         | Ours   | 72.97 | **100.00** | **84.82** | 8.60  | **99.99**  | **73.67** | **93.82** | 0.98  | $\downarrow$ **39.69**    |
> > > | Unrelated    | BD     | **73.80** | **100.00** | 6.83  | 1.50  | 92.25  | 6.25  | 12.58 | 11.42 | $\downarrow$ 78.19    |
> > > |         | EW     | 73.57 | **100.00** | 27.67 | 3.42  | 93.33  | 18.25 | 17.75 | 40.25 | $\downarrow$ 66.56    |
> > > |         | CW     | 73.45 | 99.83  | 0.25  | 1.08  | 41.08  | 4.08  | 7.67  | 0.58  | $\downarrow$ 90.71    |
> > > |         | Ours   | 72.27 | **100.00** | **97.42** | **44.67** | **100.00** | **94.08** | **97.25** | **45.17** | $\downarrow$ **20.24**    |
> > >
> > >
> > > ---

---

> > > > ### Author Response · Authors · 2022-11-13
> > > > **Author Response (Part IV)**
> > > >
> > > > **Q6**: It is unclear for how to distinguish the proposed method from traditional adversarial training method. The SOTA adversarial training methods can still be applied in the context of the watermark of deep neural networks.
> > > >
> > > > **R6**: Indeed, our approach looks like traditional adversarial training (AT). It is most probably because AT and our formulation are all min-max-based. However, **there are still many fundamental differences between our method and traditional AT**, as follows:
> > > >
> > > > - **Traditional AT and our method have different purposes**. Traditional AT aims to improve the robustness against adversarial attacks, whereas our method intends to improve the resistance to watermark removal attacks.
> > > > - **Traditional AT and our method work in different spaces**. During training, traditional AT adversarially perturb the input, whereas our method adversarially perturb model parameters.
> > > > - **Empirically, our method is significantly better than traditional AT in improving watermark robustness**. Specifically, we conduct additional experiments by performing traditional AT on watermark samples (instead of all samples to preserve high BA). The results shown in the following table verify the effectiveness of our method again.
> > > >
> > > > | Method| BA    | WSR   | FT    | FP    | ANP   | NAD   | MCR   | NNL   | AvgDrop  |
> > > > |-----|-------|-------|-------|-------|-------|-------|-------|-------|----------|
> > > > |  Vanilla  | 93.64 | 99.63 | 39.91 | 66.46 | 38.78 | 23.76 | 20.38 | 13.59 |  $\downarrow$ 65.81    |
> > > > |  AT  | 93.69 | **99.93** | 74.84 | 73.85 | 33.21 | 24.05 | 27.77 | 8.94 |  $\downarrow$ 59.48    |
> > > > |  Ours  | **93.79** | 99.92 | **98.72** | **98.64** | **99.43** | **75.87** | **78.89** | **26.12** |  $\downarrow$ **20.31**    |
> > > >
> > > > ---

---

> ### Author Response · Authors · 2022-11-16
> **Thanks to Reviewer y4Xt**
>
> We sincerely thank you again for reviewing our work and the valuable feedback, and in particular for recognizing the strengths of our paper in terms of *research significance*, *extensive experiments*, and *method effectiveness*.
>
> We realize that our previous submission may cause you some misunderstandings. We are deeply sorry about it. Please kindly let us know if you have any additional questions or require further clarification of our responses. We are happy to address them before the rebuttal ends.

---

> ### Author Response · Authors · 2022-11-30
> **A Gentle Reminder of the Final Feedback**
>
> Thank you very much again for your comments. They are extremely valuable for improving our work. We shall be grateful if you can have a look at our response and modifications, and kindly let us know if you have any other questions.

---

> ### Author Response · Authors · 2022-12-03
> **A Second Reminder of the Post-rebuttal Feedback**
>
> Dear Reviewer y4Xt,
>
> We greatly appreciate your comments. We totally understand that you may be extremely busy at this time. But we still hope that you could have a quick look at our responses to your concerns. We appreciate any feedback you could give us. We also hope that you could kindly update the rating if your questions have been addressed. We are also happy to answer any additional questions before the rebuttal ends.
>
> Best Regards,
>
> Paper4373 Authors

---

### Official Review · Reviewer_9HCi · 2022-10-25

**Confidence:** 3
**Correctness:** 3
**Technical Novelty And Significance:** 2
**Empirical Novelty And Significance:** 1
**Recommendation:** 5

**Clarity, Quality, Novelty And Reproducibility:**

The method section can be better if provided with better figures and examples. The novelty of the main approach that formulates a minimax problem seems good but the other techniques used in this approach such as cBN pose questions that whether this approach is robust enough. Some details are missing to reimplement the approach but generally seems enough to have a picture of the computation graph.

**Details Of Ethics Concerns:**

This paper works towards a better watermarked model. Adversaries can fight against this approach as well and do harm to users.

**Strength And Weaknesses:**

This language reads well. And the proposed techniques seem effective from experimental results.

However, there are several downsides:
1) Motivation is neither clear nor reasonable. One assumption made by the authors is that the adversary has methods to obtain an unauthorized copy of the watermarked model. In practice, I don't think it is possible and if that happens, one should upgrade their security system instead of finding a method that can fight against different attacks. Besides, wouldn't it be more practical to release a second watermarked model once that happened?
2) Comparison is only conducted on some baselines instead of SOTA approaches. It is mentioned in the related work (robust black-box model watermark) that several recent methods also tried to propose better methods to counter attacks. I wonder why the authors did not compare with them. Comparison with SOTA methods is desperately needed.
3) The cBN technique is a trick, lacking enough technical novelty, though it is not listed as the main contribution. Then it makes no sense to spare a huge section explaining it, especially when more details are needed to elaborate the main approach better.

**Summary Of The Paper:**

This paper proposes a minmax approach to improve the watermarked model's capacity to counter opponents' deceiving methods. This approach originates from the observation that many watermark-removed models exist around the vicinity of the watermarked one.

**Summary Of The Review:**

According to the several drawbacks I mentioned in previous parts, I do not quite believe this paper is adequate enough for publication at ICLR. The paper can be better if the motivation goes more clear and the comparison is conducted with SOTA methods.

---

> ### Author Response · Authors · 2022-11-13
> **Author Response (Part I)**
>
>
> We sincerely thank you for your valuable time and comments. We are encouraged by your positive comments on our **method effectiveness** and **paper writing**! We are deeply sorry for the misunderstandings that our paper may cause you. We will alleviate your remaining concerns, as follows:
>
>
> ---
> **Q1**: Motivation is neither clear nor reasonable. One assumption made by the authors is that the adversary has methods to obtain an unauthorized copy of the watermarked model. In practice, I don't think it is possible and if that happens, one should upgrade their security system instead of finding a method that can fight against different attacks. Besides, wouldn't it be more practical to release a second watermarked model once that happened?
>
>
> **R1:** Thank you for your question and we do understand your concerns. Security systems really matter for cloud-deployed DNNs with APIs for service. Instead, we want to note that there are also other important and practical scenarios where the model is directly exposed to adversaries. In these cases, the adversaries may remove the potential model watermark to use or release the modified model without authorization. For example,
> 1. Models are open-sourced and permitted to download for limited usage ($e.g.$, Facebook applies [OPT-175-license](https://github.com/facebookresearch/metaseq/blob/main/projects/OPT/MODEL_LICENSE.md) to their Open Pre-trained Transformers ([OPT](https://github.com/facebookresearch/metaseq)), which only permits model usage for research purposes).
> 2. Models are directly sold or bought, where the purchaser is not allowed to distribute or give the model to a third party without authorization.
> 3. Models are embedded in devices ($e.g.$, [AWS DeepLens](https://aws.amazon.com/deeplens/)), and the devices are sold or bought.
>
> In these cases, once the model copy is leaked or abused in unauthorized ways, the functionality and intellectual property embodied in it are violated, and simply replacing the watermarked model cannot recover the loss. We hope our method can better detect such violations with robust watermarks, for follow-up actions ($e.g.$, accountability, negotiation, or lawsuit).
>
>
> ---
>
> **Q2**: Comparison is only conducted on some baselines instead of SOTA approaches. It is mentioned in the related work (robust black-box model watermark) that several recent methods also tried to propose better methods to counter attacks. I wonder why the authors did not compare with them. Comparison with SOTA methods is desperately needed.
>
> **R2**: In fact, **we compared our method with 2 SOTA methods (Namba & Sakuma, 2019; Bansal et al., 2022), out of 4 methods in total mentioned in Related Works**. These two methods and ours have a similar threat model. Unfortunately, we forgot to clearly refer them. We have added more description in our experimental section in the reversion.
>
> For the remaining methods, we did not compare our method with (Li et al., 2019) since it required defenders to query the model with extremely large pixel values ($e.g.$, 500,000) that far exceed the allowed maximum value of natural images ($i.e.$, 255). **This defense can be easily circumvented by clipping image pixels to [0, 255] or refusing the predictions of these abnormal samples**. More importantly, **this defense is only feasible for models without NormLayer (e.g., BatchNorm and LayerNorm)**. As shown in the following table, training VGG-16 with BatchNorm with this method will lead to very low benign accuracy on CIFAR-10 dataset. This failure is mostly because the batch statistics (mean & variance) calculated inside BatchNorm are unduly affected by outliers caused by these extreme values in input pixels.
>
> |Model|BA|WSR|
> |-----|--|---|
> |VGG-16 (w/o BN)| 89.35 | 100.00|
> |VGG-16 (w/ BN)| **10.00** | 100.00|
>
>
> Besides, we did not compare our method with CAE (Lukas et al., 2020) since it required training 36 different models for ensemble-based defense. **This approach is very time-consuming, requiring a large amount of training resources**. We understand your concern and also compare our method with it as follows:
>
> | Method| BA    | WSR   | FT    | FP    | ANP   | NAD   | MCR   | NNL   | AvgDrop  |
> |-----|-------|-------|-------|-------|-------|-------|-------|-------|----------|
> |  Vanilla  | 93.64 | 99.63 | 39.91 | 66.46 | 38.78 | 23.76 | 20.38 | 13.59 |  $\downarrow$ 65.81    |
> |  CAE  | **93.80** | **100.00** | 67.67 | 67.67 | 61.33 | **78.67** | 68.33 | 23.33 |  $\downarrow$ 38.83    |
> |  Ours  | 93.79 | 99.92 | **98.72** | **98.64** | **99.43** | 75.87 | **78.89** | **26.12** |  $\downarrow$ **20.31**    |
>
> The aforementioned results show that our method is still (significantly) better than it in most cases (5 out of 6), verifying the effectiveness of our method.
>
> ---

---

> > ### Author Response · Authors · 2022-11-13
> > **Author Response (Part II)**
> >
> >
> >
> > ---
> > **Q3**: The cBN technique is a trick, lacking enough technical novelty, though it is not listed as the main contribution. Then it makes no sense to spare a huge section explaining it, especially when more details are needed to elaborate the main approach better.
> >
> > **R3**: Thank you for this comment. However, we have to respectfully disagree with your opinion that cBN is only a trick for several reasons.
> > - **cBN is an essential and indispensable component of our method**. We observe that the distribution of watermark samples and clean samples vary differently in Figure 2. Usually, the adversaries use only benign samples to remove the model watermark. Accordingly, using watermark samples to estimate the BN will lead to low robustness in some cases. In other words, our cBN is well-motivated and useful for our method.
> > - **cBN itself has mild effects in improving the watermark robustness** (as shown in the following table). In other words, the main benefits of our method come from the min-max-based formulation instead of the cBN.
> >
> > | APP | CBN  | BA |WSR   | FT    | FP    | ANP   | NAD   | MCR   | NNL   | AvgDrop  |
> > |-----|-----|-------|-------|-------|-------|-------|-------|-------|-------|---|
> > |    |    | 93.64 | 99.63 | 39.91 | 66.46 | 38.78 | 23.76 | 20.38 | 13.59 |  $\downarrow$ 65.81    |
> > |    | $\checkmark$   | **93.81** | 99.74 | 53.63 | 78.46 | 13.27 | 22.67 | 13.82 | 20.47 |  $\downarrow$ 65.97    |
> > | $\checkmark$   |    | 93.28 | 99.69 | 58.93 | 64.07 | 88.61 | **86.40** | 64.94 | 11.83 |  $\downarrow$ 37.23    |
> > | $\checkmark$   | $\checkmark$   | 93.79 | **99.92** | **98.72** | **98.64** | **99.43** | 75.87 | **78.89** | **26.12** |  $\downarrow$ **20.31**    |
> >
> > ---

---

> ### Author Response · Authors · 2022-11-16
> **Thanks to Reviewer 9HCi**
>
> We sincerely thank you again for reviewing our work and the valuable feedback, and in particular for recognizing the strengths of our paper in terms of *method effectiveness* and *paper writing*.
>
> We realize that our previous submission may cause you some misunderstandings. We are deeply sorry about it. Please kindly let us know if you have any additional questions or require further clarification of *our motivation*, *the comparison to SOTA defenses*, and *cBN issues*. We are happy to address them before the rebuttal ends.

---

> ### Author Response · Authors · 2022-11-30
> **A Gentle Reminder of the Final Feedback**
>
> Thank you very much again for your initial comments. They are extremely valuable for improving our work. We shall be grateful if you can have a look at our response and modifications, and kindly let us know if you have any other questions.

---

> ### Author Response · Authors · 2022-12-03
> **A Second Reminder of the Post-rebuttal Feedback**
>
> Dear Reviewer 9HCi,
>
> We greatly appreciate your initial comments. We totally understand that you may be extremely busy at this time. But we still hope that you could have a quick look at our responses to your concerns. We appreciate any feedback you could give us. We also hope that you could kindly update the rating if your questions have been addressed. We are also happy to answer any additional questions before the rebuttal ends.
>
> Best Regards,
>
> Paper4373 Authors

---

### Official Review · Reviewer_6TWD · 2022-10-26

**Confidence:** 3
**Clarity, Quality, Novelty And Reproducibility:** The paper is well-written and easy to…
**Correctness:** 4
**Technical Novelty And Significance:** 3
**Empirical Novelty And Significance:** 3
**Recommendation:** 8

**Strength And Weaknesses:**

Strength:
Watermarking neural networks is an important research problem. The "black-box" setting under which the authors studies the problem is realistic for practical applications. The proposed min-max formulation is clear and intuitive, and the experiment results support the claims that the method is effective in defending against adversarial attacks.

Weakness:
Some minor issues with writing and experiments.

1. The BatchNorm on clean samples technique should be highlighted more in Algorithm 1 as it is critical to the method.
2. Section 4.1 claims experiments were done on CIFAR-100, but I don't see any results on CIFAR-100. In particular, I am curious to see whether this method scales reasonably with respect to the number of class labels.

**Summary Of The Paper:**

This paper proposes a method for watermarking deep neural networks for classification against adversarial attacks. The authors proposed a min-max formulation of the watermark defense problem and solved the problem through first-order approximations to the inner maximization problems. Experiment results show the method is robust towards various attacks while keeping a reasonable accuracy on regular samples.

**Summary Of The Review:**

Overall, I recommend the acceptance of this paper.

---

> ### Author Response · Authors · 2022-11-13
> **Author Response**
>
> We sincerely thank you for your valuable time and comments. We are encouraged by your positive comments on our **research significance**, **realistic black-box setting**, **clear method formulation**, **method effectiveness**, and **paper writing**! We are deeply sorry for the misunderstandings that our paper may cause you. We will alleviate your remaining concerns, as follows:
>
> **Note**: All modified contents are marked in orange in our revision.
>
> ---
> **Q1**: The BatchNorm on clean samples technique should be highlighted more in Algorithm 1 as it is critical to the method.
>
> **R1**: We have updated Algorithm 1 to highlight the use of clean samples in our method. Specifically, we replace the
> $\nabla_\theta\mathcal{L}(\theta + \delta,\mathcal{B}_w)$ with $\nabla_\theta \mathcal{L}(\theta + \delta, \mathcal{B}_w;\mathcal{B}_c)$ to explicitly express the usage of clean samples. We have also added some annotations in Algorithm 1 to provide more details.
>
>
> ---
>
> **Q2**: Section 4.1 claims experiments were done on CIFAR-100, but I don't see any results on CIFAR-100. In particular, I am curious to see whether this method scales reasonably with respect to the number of class labels.
>
> **R2**: We are deeply sorry that our previous submission may cause some misunderstandings for unclear references. The detailed results on CIFAR-100 were presented in Appendix B.6. (Table 4). We also add more discussion about these results in Section 4.2 of the reversion. For your convenience, we include this table as follows:
>
> | Type    | Method | BA    | WSR    | FT    | FP    | ANP    | NAD   | MCR   | NNL   | AvgDrop  |
> |---------|--------|-------|--------|-------|-------|--------|-------|-------|-------|----------|
> | Content | BD     | **74.09** | 98.51  | 32.32 | 1.57  | 87.69  | 1.85  | 18.67 | 0.45  | $\downarrow$ 74.75    |
> |         | EW     | 73.75 | 98.21  | 18.63 | 1.44  | 88.51  | 0.79  | 2.28  | 2.52  | $\downarrow$ 79.18    |
> |         | CW     | 73.75 | 99.08  | 8.57  | 0.18  | 66.42  | 1.38  | 6.14  | 0.17  | $\downarrow$ 85.27    |
> |         | Ours   | 73.69 | **99.62**  | **97.74** | **97.25** | **99.39**  | **93.50** | **97.04** | **20.02** | $\downarrow$ **15.46**    |
> | | | | | | | | | | | |
> | Noise   | BD     | **74.13** | 99.94  | 60.54 | **10.03** | 96.55  | 20.57 | 52.77 | 0.12  | $\downarrow$ 59.85    |
> |         | EW     | 73.43 | 99.87  | 10.73 | 9.79  | 95.62  | 6.69  | 8.75  | **12.99** | $\downarrow$ 75.78    |
> |         | CW     | 73.49 | 99.98  | 24.38 | 1.80  | 55.95  | 3.28  | 38.44 | 0.05  | $\downarrow$ 79.33    |
> |         | Ours   | 72.97 | **100.00** | **84.82** | 8.60  | **99.99**  | **73.67** | **93.82** | 0.98  | $\downarrow$ **39.69**    |
> | | | | | | | | | | | |
> | Unrelated    | BD     | **73.80** | **100.00** | 6.83  | 1.50  | 92.25  | 6.25  | 12.58 | 11.42 | $\downarrow$ 78.19    |
> |         | EW     | 73.57 | **100.00** | 27.67 | 3.42  | 93.33  | 18.25 | 17.75 | 40.25 | $\downarrow$ 66.56    |
> |         | CW     | 73.45 | 99.83  | 0.25  | 1.08  | 41.08  | 4.08  | 7.67  | 0.58  | $\downarrow$ 90.71    |
> |         | Ours   | 72.27 | **100.00** | **97.42** | **44.67** | **100.00** | **94.08** | **97.25** | **45.17** | $\downarrow$ **20.24**    |
>
> The aforementioned results show that our method is still significantly better than all baseline defenses on the CIFAR-100 dataset. These results indicate that our method scales reasonably with respect to the number of classes.
>
> ---

---

> ### Author Response · Authors · 2022-11-16
> **Thanks to Reviewer 6TWD**
>
> We sincerely thank you again for reviewing our work and the valuable feedback, and in particular for recognizing the strengths of our paper in terms of *research significance*, *realistic black-box setting*, *clear method formulation*, *method effectiveness*, and *paper writing*.
>
> Please kindly let us know if you have any additional questions or require further clarification of *the details of cBN* and *the results on CIFAR-100*. We are happy to address them before the rebuttal ends.

---

> ### Author Response · Authors · 2022-11-30
> **A Gentle Reminder of the Final Feedback**
>
> Thank you very much again for your initial comments. They are extremely valuable for improving our work. We shall be grateful if you can have a look at our response and modifications, and kindly let us know if you have any other questions.

---

> ### Author Response · Authors · 2022-12-03
> **A Second Reminder of the Post-rebuttal Feedback**
>
> Dear Reviewer 6TWD,
>
> We greatly appreciate your initial comments. We totally understand that you may be extremely busy at this time. But we still hope that you could have a quick look at our responses to your concerns. We appreciate any feedback you could give us. We also hope that you could kindly update the rating if your questions have been addressed. We are also happy to answer any additional questions before the rebuttal ends.
>
> Best Regards,
>
> Paper4373 Authors

---

### Decision · Program_Chairs · 2023-01-20

**Decision:**

Reject

**Justification For Why Not Higher Score:**

There are a few concerns from reviewers on the motivation, novelty, problem formulation, which are not fully addressed by the authors.

**Justification For Why Not Lower Score:**

N/A

**Metareview: Summary, Strengths And Weaknesses:**

Summary:
Based on the observation that many watermark-removed models are in the vicinity of the watermarked one in parametric space, the authors propose an adversarial training-based framework to ensure robustness in the watermark of deep neural networks.

Strength:
1. The proposed technique is clear and seems effective from experimental results.
2. The research problem is important as the paper validates the vulnerability in the watermark of deep neural networks and shows that it is necessary to develop a robust watermark framework.

Weakness:
There are a few concerns from reviewers on the motivation, novelty, problem formulation.